# PSIP1/LEDGF reduces R-loops at transcription sites to maintain genome integrity

Sundarraj Jayakumar[1,2], Manthan Patel[1], Fanny Boulet[1], Hadicha Aziz[1], Greg N. Brooke [3], Hemanth Tummala [1] & Madapura M. Pradeepa [1] ✉

R-loops that accumulate at transcription sites pose a persistent threat to genome integrity. PSIP1 is a chromatin protein associated with transcriptional elongation complex, possesses histone chaperone activity, and is implicated in recruiting RNA processing and DNA repair factors to transcription sites. Here, we show that PSIP1 interacts with R-loops and other proteins involved in R-loop homeostasis, including PARP1. Genome-wide mapping of PSIP1, R-loops and γ-H2AX in PSIP1-depleted human and mouse cell lines revealed an accumulation of R-loops and DNA damage at gene promoters in the absence of PSIP1. R-loop accumulation causes local transcriptional arrest and transcription-replication conflict, leading to DNA damage. PSIP1 depletion increases 53BP1 foci and reduces RAD51 foci, suggesting altered DNA repair choice. Furthermore, PSIP1 depletion increases the sensitivity of cancer cells to PARP1 inhibitors and DNA-damaging agents that induce R-loop-induced DNA damage. These findings provide insights into the mechanism through which PSIP1 maintains genome integrity at the site of transcription.

The newly synthesised nascent RNA can bind to the template DNA during transcription, forming an RNA-DNA hybrid. This hybrid structure combined with the displaced single-stranded DNA is termed R-loops[1]. Physiological R-loops play an essential regulatory role in DNA replication, DNA repair, transcription initiation and termination, and many other cellular processes[2]. On the other hand, accumulating unscheduled R-loops at the site of transcription is known as pathological R-loops. The accumulation of these unscheduled R-loops leads to RNA polymerase-II (RNAPII) pausing and transcriptional arrest[3]. R-loops also cause collision between stalled transcription and replication forks, leading to DNA damage and increased genomic instability[4]. To overcome this, several RNA processing factors (SRSF1), nucleases (RNASEH1 and RNASEH2) and helicases (SETX, DDX1, DDX19 etc.) reduce R-loop levels[2]. Dysregulation of R-loops is linked to many neurological conditions, autoimmune disorders and cancers (reviewed in ref. 5). Intriguingly, R-loops are also accumulated due to DNA double-stranded breaks (DSBs) at the transcription sites. R-loops have also been shown to facilitate DNA repair;

however, the mechanism through which R-loops feedback to transcription and DSB repair choice is unclear (reviewed in ref. 6).

PC4 and SF2 interacting protein (PSIP), also known as LEDGF, is a multifunctional chromatin protein, a proto-oncogene that promotes cell survival, cancer cell proliferation[7–9] and chemotherapy resistance[8,10,11]. PSIP1 binds to methylated histone H3 lysine 36 (H3K36me) via the PWWP domain, and PSIP1 binding is enriched at the sites of RNAPII transcription[12,13]. *PSIP1* encodes two protein isoforms—a shorter PSIP/p52 and a longer PSIP/p75—generated by alternative splicing (Fig. 1a). The smaller PSIP1/p52 isoform binds to RNA processing proteins and modulates alternative splicing[13]. In contrast, PSIP1/p75 guides the integration of HIV to expressed gene bodies via the interaction of HIV integrase with integrase binding domain (IBD)[14]. PSIP1/75 also interacts with the transcription elongation complex via IBD, which is homologous to the TFIIS N-terminal domain (TFIIS-NTD) (Fig. 1a)[15]. Increased RNAPII pausing and backtracking due to TFIIS mutations results in R-loop formation, leading

[1]Blizard Institute; Faculty of Medicine and Dentistry, Queen Mary University of London, London, UK. [2]Bhabha Atomic Research Centre, Mumbai, India. [3]School of Life Sciences, University of Essex, Colchester, UK. ✉e-mail: p.m.madapura@qmul.ac.uk

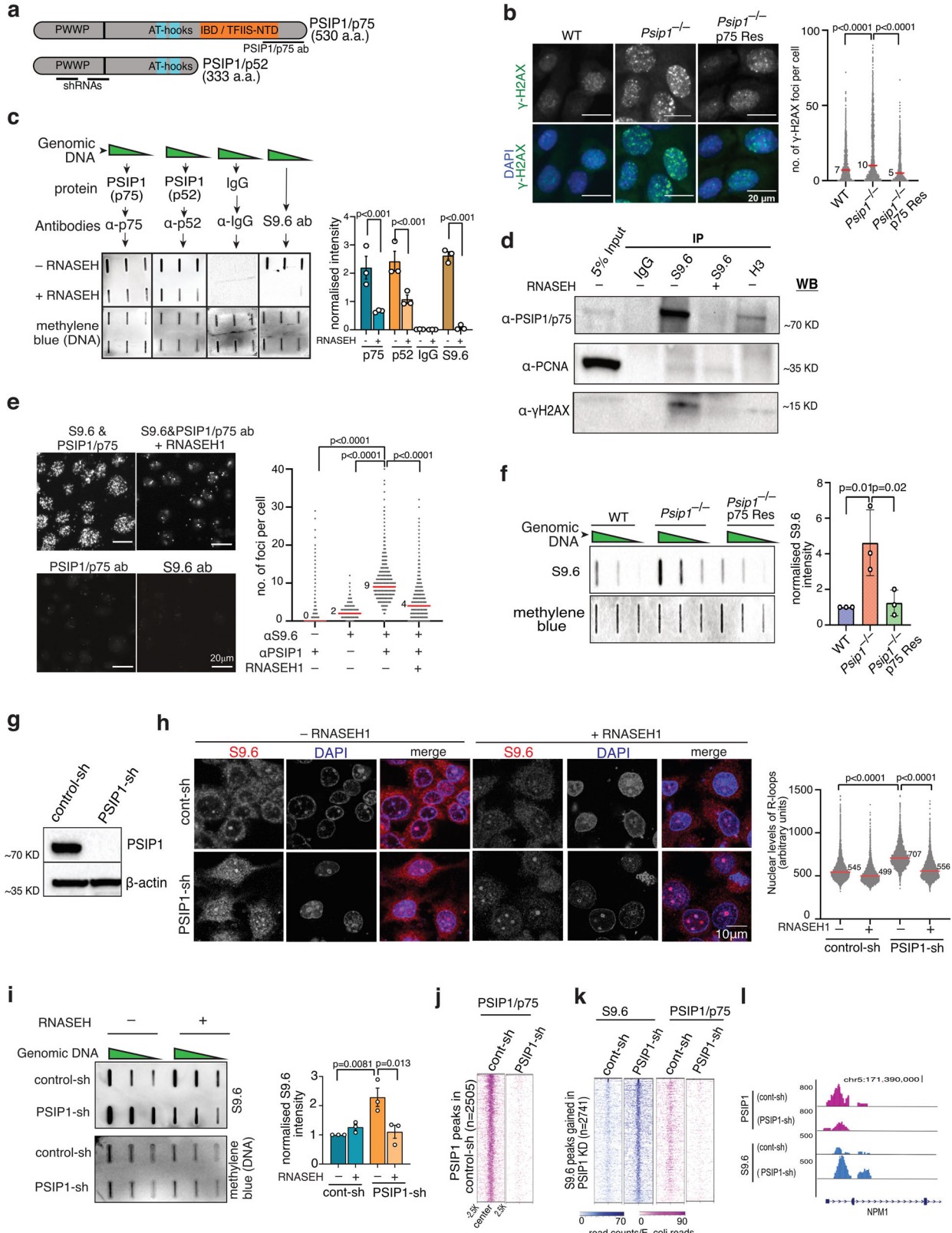

to genome instability, directly linking transcription stress, R-loop formation, and DNA damage[16]. Interestingly, PSIP1 aids in transcription elongation by functioning as a histone chaperone in the absence of the FACT complex in differentiated cells[17]. It also recruits CtIP to DNA damage sites to promote homologous recombination (HR) at transcribing regions[18,19].

Here, we discover that PSIP1 interacts with the R-loop directly, and several proteins, including PARP1, that are involved in R-loop homeostasis. We also demonstrate that PSIP1 reduces R-loop levels at the site of RNAPII transcription to maintain genome stability and promotes the repair of DNA damage induced by R-loops by homologous recombination (HR). Furthermore, PSIP1 depletion sensitises cancer

**Fig. 1 | PSIP1 depletion leads to the accumulation of R-loops. a** Schematic showing PSIP1 domains. **b** Immunofluorescence images and γ-H2AX foci in wild-type, *Psip1*⁻/⁻ and *Psip1*⁻/⁻ p75R MEFs (*n* > 1500 nuclei, three independent experiments; median values in red bar; p-values obtained using two-tailed Mann–Whitney test). **c** Slot-blot images showing the binding of p75/p52 isoforms of PSIP1 with RNA-DNA hybrids. RNASEH-treated genomic DNA and IgG served as a control. Normalised band intensity (right) with mean ± SD (*n* = 3 independent experiments; *p*-values by one-way ANOVA followed by Tukey's test). **d** Immunoblots of S9.6, H3 and IgG immunoprecipitated lysates from RWPE-1 nuclear extract with indicated antibodies. **e** Representative images and dot-plot showing PLA foci between S9.6 and PSIP1 in control and RNASEH1 overexpressed RWPE-1 cells (*n* > 1400 nuclei observed over three independent experiments; median values indicated with a red line; *p*-values obtained using two-tailed Mann–Whitney test). **f** Slot-blot for R-loop using S9.6 antibody from cells as in **b**. The normalised band intensity (right) was plotted as mean ± SD (*n* = 3 independent experiments; *p*-values by one-way ANOVA

followed by Tukey's test). **g** Immunoblot with PSIP1/p75 and β-actin for lysate from PSIP1-shRNA (PSIP1-sh) and non-targeting control shRNA (control-sh). **h** S9.6 immunofluorescence in RWPE-1 cells (left) and quantification (representative images; *n* > 4000 nuclei from three independent experiments; median values are indicated with the red line; *p*-values by two-tailed Mann–Whitney test). **i** Like **f**, but for PSIP1 knockdown (PSIP1-sh) and control knockdown (control-sh) RWPE-1 cells. R-loops isolated from cells overexpressing RNASEH1 also was used for slot-blot. The normalised band intensity has been plotted as mean ± SD (*n* = 3 independent experiments; *p*-values by one-way ANOVA followed by Tukey's test). **j** Heatmaps showing E. coli spike-in normalised CUT&Tag reads for PSIP1/p75 in control and PSIP1 knockdown RWPE-1 cells. **k** Heatmap showing R-loop and PSIP1/p75 levels in PSIP1-KD and control RWPE-1 cells. **l** Genome-browser track (hg38) showing the CUT&Tag signal (read counts) for PSIP1 and R-loop in control and PSIP1-KD RWPE-1 cells. Source data are provided as a Source Data file.

cells to clastogens that cause R-loop-induced DNA damage and PARP1 inhibitors.

## Results

### PSIP1 interacts with R-loops, and PSIP1 depletion elevates the R-loop level

We aimed to investigate the mechanism through which PSIP1 maintains genomic stability. Consistent with the known function of PSIP1 in promoting HDR[19,20], *Psip1*⁻/⁻ mouse embryonic fibroblast (MEFs) displayed an elevated number of γ-H2AX foci, a marker of DNA damage (Fig. 1b and Supplementary Fig. 1a). This increase in DNA damage could be reversed by re-expression of PSIP1/p75, indicating a role of PSIP1/p75 in maintaining genomic integrity in the cells (Fig. 1b). Published proteomics data from immunoprecipitation (IP) of endogenous PSIP1/LEDGF identified several proteins involved in transcription, RNA processing and DNA repair[13,21,22]. Forty-six proteins enriched in the PSIP1 IP overlap with the previously reported S9.6 antibody-based R-loop interactome[23]. These overlapping proteins include DNA-RNA helicases, splicing and DNA repair factors (Supplementary Fig. 1b). PSIP1 is also enriched among other R-loop modulating proteins in the RNASEH1 TurboID-based proximity-dependent labelling system[24]. Since PSIP1 interacts with RNAPII transcriptional elongation machinery[15] and is associated with transcriptionally active chromatin[13,21], we hypothesised that PSIP1 could prevent R-loop accumulation or promote the resolution of unscheduled R-loops at the sites of RNAPII transcription. We, therefore, sought to investigate whether PSIP1 is involved in R-loop homoeostasis.

Firstly, we evaluated the ability of PSIP1 to bind R-loops in vitro by incubating the genomic DNA blotted membrane with recombinant PSIP1/p52 and PSIP1/p75, followed by the detection of PSIP1 isoforms using antibodies. Both p75 and p52 isoforms showed binding to R-loops, which was reduced upon RNASEH treatment (Fig. 1c). IP with the S9.6 monoclonal antibody that binds RNA-DNA hybrids (R-loops)[25], detected the interaction of PSIP1/p75 with the R-loop complex. This interaction was abolished upon degradation of R-loops in the RNASEH-treated extract (Fig. 1d). Proximity ligation assay (PLA) using S9.6 and PSIP1/p75 antibodies confirmed the interaction of PSIP1/p75 with R-loops (Fig. 1e). Quantifying bulk R-loop levels using immunoblotting of genomic DNA using the S9.6 antibody showed a stark increase in R-loop levels in *Psip1*⁻/⁻ MEFs compared to the wild type (Fig. 1f). This was rescued upon PSIP1/p75 re-expression, implying the role of PSIP1 in reducing the R-loop level (Fig. 1f). The elevated R-loop levels in *Psip1*⁻/⁻ MEFs were sensitive to RNASEH treatment, confirming the specificity of this R-loop detection method (Supplementary Fig. 1c).

Using two independent short-hairpin (sh) RNAs that target both isoforms of PSIP1, we generated normal prostate epithelial (RWPE-1) and HEK293T cell lines depleted of PSIP1 (PSIP1-KD) (Fig. 1g; Supplementary Fig. 1d and Supplementary Fig. 1g). Immunofluorescence imaging followed by a high-content microscopy and slot-blot analysis

revealed a significantly higher level of R-loops in the PSIP1-KD compared to the control RWPE-1 nuclei (Fig. 1h, i) and in HEK293T cells (Supplementary Fig. 2a). RNASEH1 overexpression in these cells led to the reversal of R-loop accumulation (Fig. 1h, i and Supplementary Fig. 2a). A similar increase in R-loop intensity was observed when PSIP1 was depleted using an independent shRNA in RWPE-1 cells and HEK293T cells (Supplementary Fig. 1d–i). Similarly, this increase in R-loop levels mediated by PSIP1 depletion could be rescued by overexpression of p75 or p52 isoforms (Supplementary Fig. 2b).

To further investigate whether R-loop accumulation is directly due to PSIP1 depletion, we performed two replicates of Cleavage Under Targets and Tagmentation (CUT&Tag)[26], using PSIP1/p75, S9.6 and γ-H2AX antibodies. Reduction in PSIP1/p75 signal in PSIP1-KD cells confirms the specificity of the PSIP1/p75 CUT&Tag data (Fig. 1j). Consistent with the slot-blot and immunofluorescence data, we found a nearly five-fold increase in R-loop peaks (*n* = 2741) in PSIP1-KD RWPE-1 cells compared to the control (*n* = 545). Notably, the regions that gained R-loops upon PSIP1 depletion are occupied by PSIP1/p75 in control cells (Fig. 1k, l).

### PSIP1 depletion leads to increased DNA damage at R-loop sites

The accumulation of unscheduled R-loops is known to cause DNA replication stress, activating DNA damage response pathways[4]. We found an elevated γ-H2AX level in PSIP1-KD RWPE-1 (Fig. 2a and Supplementary Fig. 3b) and HEK293T cells (Supplementary Fig. 3a) compared to the control, further confirming the role of PSIP1 in reducing DNA damage. The number of γ-H2AX foci was also significantly higher in PSIP1-depleted RWPE-1 cells (Fig. 2b). Notably, the increased γ-H2AX foci were reversed upon RNASEH1 overexpression (Fig. 2b). 53BP1 is crucial to DNA double-strand breaks (DSB) signalling and repair; the increase in 53BP1 foci and bulk level upon PSIP1 depletion shows increased DNA damage signalling in the absence of PSIP1, which could be rescued by overexpression of RNASEH1 (Fig. 2c, d). RNAseq data analysis showed minimal changes in the expression of 186 genes (Supplementary Data 1) implicated in R-loop homoeostasis (Supplementary Fig. 2c)[27], indicating that the elevated R-loop levels are due to the PSIP1 depletion but not the indirect consequence of the altered transcriptional programme.

CUT&Tag with γ-H2AX antibodies revealed an increase in the number of γ-H2AX peaks in PSIP1-KD compared to control (Fig. 2e). γ-H2AX peaks gained in PSIP1-KD also show an increased R-loop signal (Fig. 2e, f and Supplementary Fig. 3c). Notably, the elevated S9.6 and γ-H2AX levels in PSIP1-KD could be rescued by RNASEH1 overexpression (Fig. 2e, f), confirming the increased DNA damage signalling in PSIP1-KD is due to elevated R-loops. CUT&Tag data is consistent with the increased bulk γ-H2AX level and the number of γ-H2AX foci in the PSIP1-KD (Fig. 2a, b). Furthermore, PLA analysis revealed an increased number and co-localisation of γ-H2AX and R-loop foci in PSIP1-KD, further confirming that R-loops accumulated

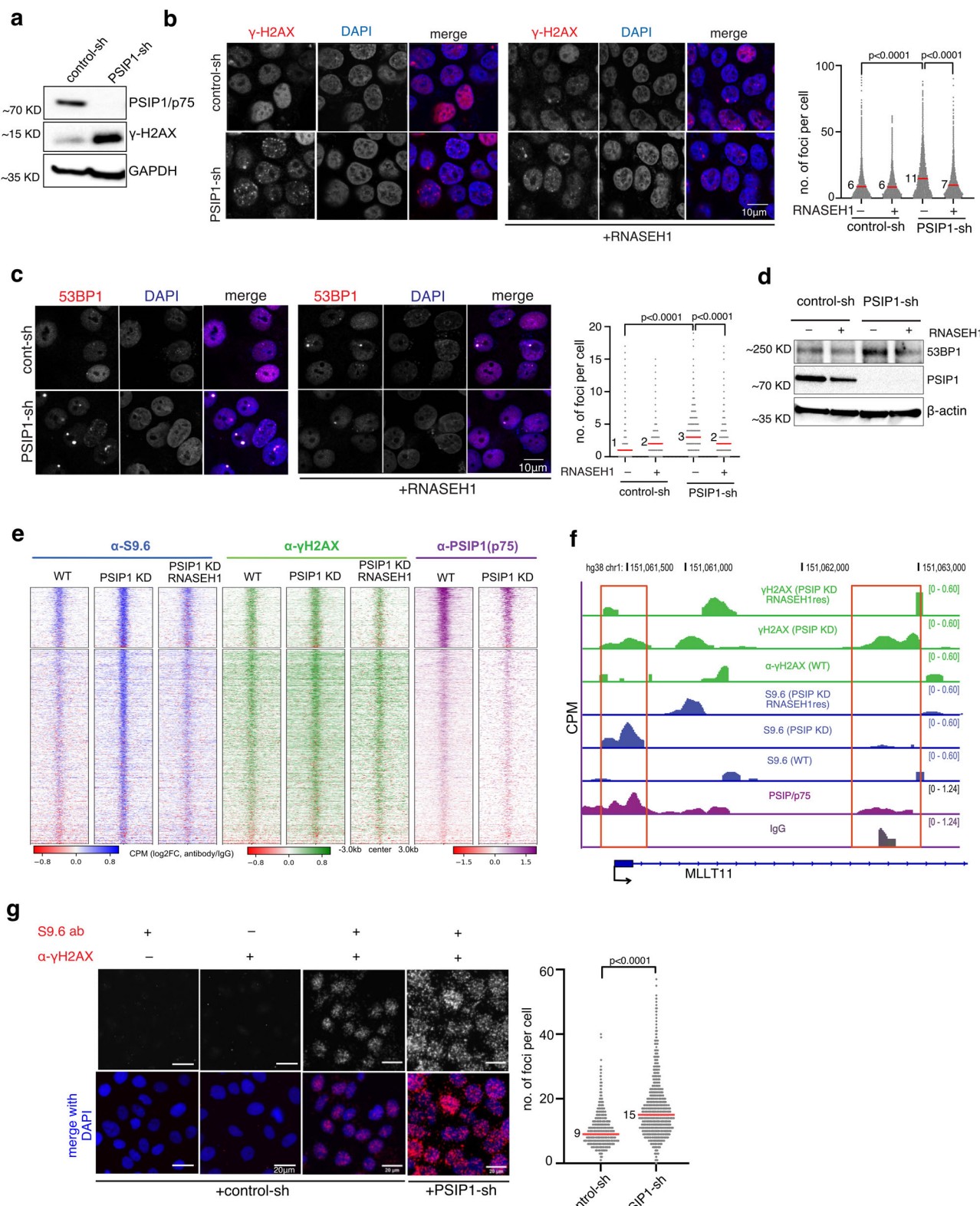

upon PSIP1 depletion are responsible for increased DNA damage in PSIP1-KD (Fig. 2g).

## R-loops cause transcriptional arrest and transcription-replication conflict

R-loop peaks gained, as a result of PSIP1-KD, were enriched around promoter or transcription start sites (TSS) and gene bodies compared to intergenic regions (Fig. 3a, b). Similarly, PSIP1/p75 binding showed enrichment around gene promoters and gene bodies. Peak overlap analysis revealed that the 909 R-loop peaks detected in PSIP KD overlapped with PSIP1/p75 peaks in control cells. Similarly, 239 S9.6 and γ-H2AX peaks that appeared upon PSIP1 depletion overlapped with the PSIP1/p75 peaks in control (Fig. 3c). Accumulated R-loops show enrichment for G-rich promoters with high GC skew, suggesting

**Fig. 2 | PSIP1 depletion leads to R-loop-mediated DNA damage. a** Western blot showing the levels of γ-H2AX in control and PSIP1 KD RWPE-1 cells. **b** Representative IF images and dot-plot showing the γ-H2AX foci in control and PSIP1 KD RWPE-1 cells. The cells overexpressing RNASEH1 were also used as controls. The number of foci per cell was quantified and plotted as a dot-plot ($n > 8000$ nuclei from three independent experiments; Median values are indicated with a red line; $p$-values by two-tailed Mann–Whitney test). **c** Like **b** but for 53BP1 foci (representative images; $n > 3000$ cells for each group from three independent experiments; median values are indicated with red bars; $p$-values by two-tailed Mann–Whitney test). **d** Immunoblot images showing the bulk levels of 53BP1 protein in control, PSIP1 KD

and upon RNASEH1 overexpression in HEK293T cells. **e** Heatmap showing CUT&-Tag reads (normalised to *E. coli* reads) for γ-H2AX and PSIP1/p75 across S9.6 peaks gained in PSIP1-KD HEK293T cells. **f** Genome-browser tracks showing CUT&Tag data for PSIP/p75, R-loops (S9.6 ab) and γ-H2AX in control and PSIP-KD HEK293T cells. **g** Representative images and dot-plot of PLA between R-loops (S9.6) and γ-H2AX antibodies in control and PSIP-KD RWPE1 cells (representative images; $n > 1000$ nuclei observed from three independent experiments; median values indicated with red bar; $p$-values obtained using two-tailed Mann–Whitney test). Source data are provided as a Source Data file.

that PSIP1 reduces R-loops at G-rich promoters or genes regulated by strong CpG islands with high transcriptional activity (Supplementary Fig. 4a, b)[28].

R-loops can induce transcriptional termination or arrest[16,29]. To investigate whether R-loop accumulation upon PSIP1 depletion affects transcription at the sites of R-loop accumulation, we performed transient transcriptome sequencing (TT-seq) by labelling newly synthesised transcripts with 4-thiouridine (4sU)[30]. Meta-analysis of TT-seq data showed an overall reduced level of nascent transcription across the gene transcription unit. The reduction in TT-seq signal was higher at the promoter or upstream of gene TSS compared to the rest of the gene body (Fig. 3d). Replotting of TT-seq reads at R-loop peaks that are gained in PSIP KD revealed a stark reduction in the nascent transcript levels (Fig. 3d). Furthermore, triptolide-mediated transcription inhibition reversed the elevated R-loop levels in PSIP1-KD (Fig. 3e). These results suggest the role of PSIP1 in reducing the R-loop level during ongoing transcription, accumulation of these R-loops, which can otherwise lead to transcriptional arrest. However, since PSIP1 is a transcriptional regulator, it is tricky to infer the direct effect of PSIP1 depletion on R-loop mediated transcriptional arrest.

The R-loop-induced transcriptional arrest can cause collisions with the DNA replication forks, resulting in DNA damage. R-loop accumulation in the head-on orientation with the replication fork can cause a transcription-replication conflict (TRC)[4]. Hence, we monitored the interaction between the transcription and replication machinery using the PLA for RNAP II and PCNA antibodies, which mark transcription complexes and replication forks. Increased RNAPII and PCNA PLA foci in PSIP1 KD show a higher level of TRC in the absence of PSIP1; RNASEH1 overexpression in the PSIP1 KD leads to significantly reduced RNAPII and PCNA PLA foci suggesting reduced TRC due to rescue of elevated R-loops (Fig. 3f). Reduced incorporation of 5-ethynyl-2'-deoxyuridine (EdU) into DNA in PSIP1-KD in an EdU incorporation assay implies a slower replication rate upon PSIP depletion (Supplementary Fig. 5a). Cell cycle analysis revealed that reduced EdU incorporation is not due to changes in the cell cycle stages due to PSIP1 depletion (Supplementary Fig. 5b, c). These results show that PSIP1 reduces transcription-induced R-loops, which can otherwise lead to local transcriptional arrest due to collision with the replication fork, resulting in reduced replication rate.

### PSIP1 promotes the repair of DNA damage induced by transcription and R-loops

Our data showed that increased DNA damage upon PSIP1 depletion is due to elevated R-loops (Fig. 2b–g). The slow repair of R-loop-induced DNA damage could also contribute to the observed increase in DNA damage. The absence of PSIP1 could slowdown repair of R-loop mediated DNA damage, as PSIP1 is known to promote HR[19], and interacts with DNA repair factors (Supplementary Fig. 1b). The DNA damage caused by R-loops is repaired by transcription-coupled double-strand break repair[31]. To investigate the role of PSIP1 in the repair of R-loop mediated DNA damage at the site of transcription, we treated the cells with illudin-S and etoposide, which are known to cause transcription-coupled DNA damage. PSIP1 depletion led to an increased sensitivity to illudin-S mediated DNA lesions that are specifically repaired by the transcription-coupled nucleotide excision repair

(TC-NER) pathway but not global nonhomologous end joining (NHEJ) and base excision repair (BER)[32]. Similarly, PSIP1 depletion increased sensitivity to etoposide, which is known to bind topoisomerase II and block transcription and replication[33] (Fig. 4a). Phleomycin treatment, which induces DSBs showed similar sensitivity, but not aphidicolin that causes cell cycle arrest[34,35] (Fig. 4a and Supplementary Fig. 6a). This specific sensitivity to clastogens, that interfere with the transcription process, reveals the role of PSIP1 in promoting DNA repair at the site of transcription. Further, western blotting for γ-H2AX revealed a sustained DNA damage response in phleomycin-treated PSIP1-KD compared to the control cells (Fig. 4b).

Enrichment of PSIP1/p75 in γ-H2AX IP shows that PSIP1 is recruited to DNA damage sites (Fig. 4c). Notably, PSIP1 recruitment to γ-H2AX sites is further increased when treated with camptothecin (CPT), which induces R-loop accumulation (Fig. 4c and Supplementary Fig. 6c, d) by inhibiting topoisomerase 1 (TOP1) at active promoters[36]. However, PSIP1 recruitment to γ-H2AX chromatin was reduced upon overexpression of RNASEH1 (Fig. 4c). The protein level of DNA-PK, which is involved in the NHEJ pathway, is increased in PSIP1-KD cells (Fig. 4b). Similarly, the number of 53BP1 (known to promote NHEJ) foci, were significantly higher in PSIP1-KD and upon CPT treatment compared to the control (Fig. 4d, e). A similar increase in 53BP1 foci was also observed after phleomycin treatment upon PSIP1-KD (Supplementary Fig. 6b). In contrast, upon PSIP1 depletion, the number of RAD51 foci—a marker of the HR pathway, was significantly less in control and CPT treated cells (Fig. 4d, e). These results agree with the role of PSIP1 in promoting the HR repair pathway[18,19] and suggest that PSIP1 facilitates the repair of R-loop mediated DNA damage through the HR pathway at the transcription sites.

### PSIP1 interacts with PARP1 and reduced PSIP1 levels sensitise cancer cells to transcription-induced DNA damage

PARP1 is a vital DNA repair factor and also an R-loop resolving factor. PARP1 was one of the significant interactors of PSIP1/p75 and was also enriched in the R-loop proteome[22,23] (Supplementary Fig. 1b). We further confirmed the interaction of PARP1 and PSIP1 with the R-loops by S9.6 antibody IP (Fig. 5a). Reciprocal α-PSIP1-IP performed in the presence of benzonase (a universal nuclease, which degrades both RNA and DNA) (Fig. 5b), together with co-IP of HA-tagged PSIP1/p75 and PSIP1/p52 with GFP-tagged PARP1 confirmed the interaction between both PSIP1 isoforms with PARP1, along with γ-H2AX (Fig. 5c). Comparison of the PSIP1 transcripts across different cancer types in the TCGA database revealed lower PSIP1 levels compared to the respective control tissues (Fig. 5d). Cells with elevated levels of R-loops are sensitive to PARP1 inhibitor-induced cell death[37]. Hence, we aimed to investigate whether cancers with low PSIP1 levels are sensitive to PARP1 inhibitors or illudin-S that cause transcription-coupled DNA damage. Prostate cancer cell lines (LNCaP and PC3) showed lower levels of PSIP1 compared to normal prostate epithelial cells (RWPE-1) (Fig. 5e). These prostate cancer cells showed significantly higher sensitivity to the PARP1 inhibitor (PARPi) olaparib compared to RWPE-1 cells (Fig. 5f). PSIP1 depletion in LNCaP cells increased olaparib sensitivity, which could be partially rescued by RNASEH1 overexpression (Fig. 5g, h). These results show that PSIP1 and PARP1 synergistically

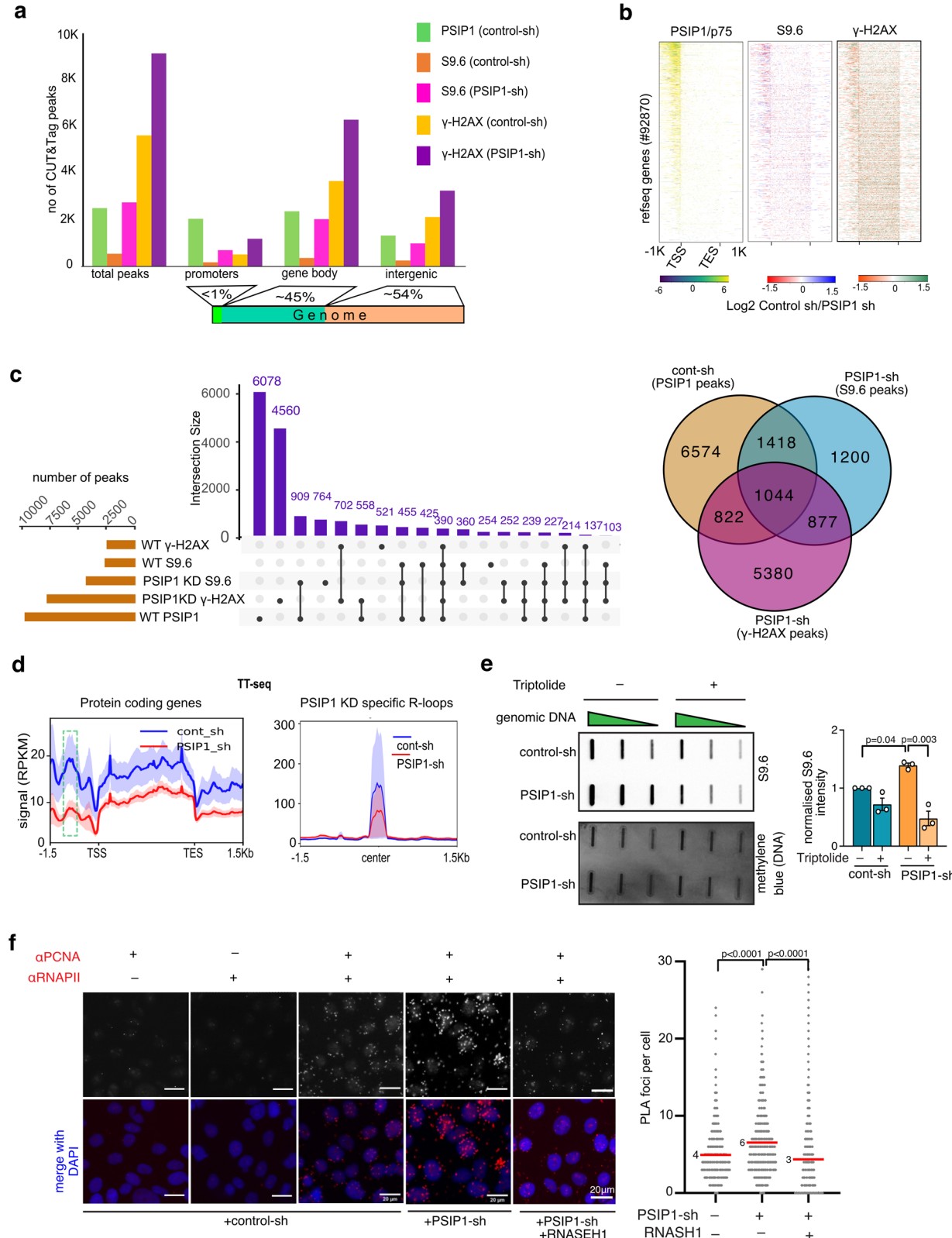

reduce R-loops-induced genomic instability. Depletion of PSIP1 also led to hypersensitivity of LNCaP cells to illudin-S, which was reversible by RNASEH1 overexpression (Fig. 5i). These data suggest that the clastogens and PARP1 inhibitors that are known to interfere with the transcription-coupled DNA repair mechanism could be effective in inducing synthetic lethality in specific cancers.

## Discussion

Accumulation of unscheduled co-transcriptional R-loops affects transcription elongation, compromises genome integrity, and is implicated in cancer and inflammation. Here, we demonstrate that PSIP1 binds to R-loops and proteins associated with R-loops, including PARP1. PSIP1 reduces R-loops formed during transcription to minimise

**Fig. 3 | R-loop accumulation at promoters leads to transcription-replication conflict. a** Distribution of PSIP1, R-loop (S9.6 ab), and γ-H2AX CUT&Tag peaks from RWPE-1 cells around the gene transcription start sites (promoters), gene bodies and intergenic regions. **b** Heatmaps showing the log2 fold change in read counts between control and PSIP1-KD CUT&Tag reads for PSIP1/p75, R-loops (S9.6 ab) and γ-H2AX in RWPE-1 cells across the NCBI reference genes. **c** upSet plot and Venn diagram (right) showing the unique and overlapping peaks obtained from CUT&Tag reads for PSIP1, S9.6 and γ-H2AX in HEK293T cells. The *x*-axis shows the number of peaks, and the Y-axis shows the number of intersections. **d** Average profile with SD (shaded regions of blue and red) of TT-seq (RPKM) across protein-coding genes (left), dotted box shows promoter region and around the centre of the S9.6 peaks gained in PSIP KD RWPE-1 cells (right). **e** Slot blot using S9.6 antibody in control and PSIP-KD RWPE-1 cells after treatment with transcriptional inhibitor (triptolide 50 nM; 36 h), methylene blue staining of the same DNA served as a loading control. Normalised intensity of S9.6 intensities was plotted as mean ± SD (*n* = 3 independent experiments; *p*-values by one-way ANOVA followed by Tukey's multiple comparison test). **f** Representative PLA image between α-PCNA and α-RNAPII antibodies obtained from control and PSIP1 KD RWPE-1 cells (left). The number of PLA foci per cell observed between PCNA and RNAPII was quantified and plotted as dot blot (*n* > 1000 cells over three independent experiments; red line shows the median value that has been indicated; *p*-values obtained from two-tailed Mann–Whitney test). Source data are provided as a Source Data file.

DNA damage and genomic instability (Fig. 5j). R-loops and γ-H2AX accumulation are sensitive to inhibition of transcription and over-expression of RNASEH1, which confirms the elevated DNA damage in PSIP1-KD cells due to unresolved R-loops that appear at the site of transcription. Furthermore, PSIP1/p75R-loops are detected at pro-moters and gene bodies, confirming the role of PSIP1 in resolving transcription-induced R-loops. We further demonstrate that elevated DNA damage is due to R-loop-induced collision between RNAPII transcription with the replication fork (Fig. 5j).

PSIP1 isoforms are enriched at the transcription unit of expressed genes through binding to di and tri-methylated H3K36[12,13]. Recently, PSIP1 have been shown to function as a histone chaperone in the absence of the FACT complex in differentiated cells to facilitate transcriptional elongation through nucleosomes[17]. Interestingly, FACT complex subunits (SSRP1 and SPT16) are detected in the R-loop and PSIP1/p75 interactome, also known to aid the R-loop resolution[22,23,38]. The intrinsic transcript-cleavage activity of RNAPII requires TFIIS to suppress R-loop formation by reducing RNAPII pausing and backtracking[16]. Intriguingly, PSIP1/p75 also has a TFIIS N-terminal domain that interacts with the transcription elongation complex[15]. These pieces of evidence suggest a broader role of histone chaperones in facilitating transcriptional elongation by reducing R-loops formed at the transcription site. Several R-loop resolving proteins such as TFIIS, PSIP1, PARP1, FACT complex, and topoisomerases are shown to function in efficient transcriptional elongation[16,23,38–41]. It is plausible that PSIP1 restrains the transcription elongation when it encounters R-loop mediated DSBs to facilitate the resolution of R-loop and promote HR-mediated DNA repair[18,42], as PSIP1 is also known to restrain transcription elongation[21].

Several proteins involved in transcription, RNA processing and DNA repairs, such as the PARP1, TFIIS, BRCA1, THO complex, SRSF1, DDX23, SETX, DHX9, TOP1 and others, are known to reduce R-loop burden and genome instability[3,43–46]. Interestingly, the splicing factor SRSF1, one of the significant interacting partners of PSIP1, also reduces the genomic instability caused by R-loops[47]. Further investigation will be needed to test whether the R-loops, elevated upon PSIP1 depletion, are due to dysregulated RNA processing due to altered recruitment of splicing factors and RNA processing[13].

The unscheduled R-loops can induce transcriptional repression via various mechanisms[48]. DNA damage-induced transcriptional arrest is essential to prevent replication and transcription collision due to aberrant transcripts, which could also generate R-loops[49]. Reduced nascent transcript levels at R-loop sites and increased PLA foci between PCNA and RNAPII show that unresolved R-loops lead to increased transcription-replication conflict (TRC). Although most PSIP1 binding and PSIP1-KD-specific R-loops are detected around promoters and gene bodies, intergenic regions also show enrichment for R-loops and γ-H2AX. Although most of the R-loops and DNA damage were detected at gene transcription units, distal regulatory elements are also a source of enhancer RNA transcription that could lead to R-loops and DSBs[50].

The interaction of PSIP1/p75 with CtIP and various DNA repair factors also suggests its role in promoting HR and improving the efficiency of DNA repair during transcription[19,22,51]. PSIP1 promotes HR in transcribing regions[18,19]. PARP1 inhibitors are successfully used for cancers deficient in HR due to *BRCA* mutations[52]. Importantly, our data show that depletion of PSIP1 results in higher sensitivity of prostate cancer cells to clastogens that cause transcription-coupled DNA damage and a PARP1 inhibitor. Although PSIP1-KD sensitised cancer cells to clastogens, including PARPi, further studies are needed to validate their efficacy in vivo. Since we detect accumulation of both R-loops and DNA damage in the absence of PSIP1, some of the R-loops accumulated in the PSIP1-KD could be due to DNA damage, as double-stranded breaks are known to generate R-loops[53].

In summary, we show that PSIP1 interacts with proteins involved in the R-loop homoeostasis, such as DNA repair, RNA processing factors, and PARP1. PSIP1 maintains genomic stability at the site of transcription by a) reducing the R-loop levels by interacting or recruiting R-loop processing proteins and b) promoting the HR repair pathway to repair damage caused by transcription-induced R-loops. Our data also revealed an increased sensitivity of cancer cells with low levels of PSIP1 to drugs that induce transcription-associated DNA damage and PARP1 inhibitors. With further in vivo validations, insights from this work could be used to therapeutically target cells with higher R-loop levels, therefore having significant implications for diseases such as cancer.

## Methods
### Materials
The materials used in the study and the details are listed in Supplementary Data 2.

### Cell culture
RWPE-1, PC-3, LNCaP and HEK293T cells were procured from the American Type Culture Collection. *Psip1⁻/⁻* and its corresponding WT MEFs were a kind gift from Prof. Alan Engelman (Dana-Farber Cancer Institute, USA)[54]. The *Psip1⁻/⁻* P75 rescue cells were generated in the lab by expressing lentiviral with P75 cDNA[55]. RWPE-1 cells were cultured in a K-SFM medium. LNCaP cells were cultured in an RPMI medium. PC-3, HEK293T and MEF cells were cultured in a DMEM medium. The medium was supplemented with 1X penicillin, streptomycin solution, and 10 % FBS. Cells are grown as an adherent monolayer culture in the appropriate medium, in the incubator at 37 °C and 5% CO₂.

### Generation of PSIP1 knock-down cell lines
Lentiviral shRNA vectors targeting PSIP1 (PSIP1 ShRNA1 TRCN0000298567, PSIP1 shRNA2 TRCN0000286345) and non-targeting control were procured from Sigma Life Sciences. Exponentially growing cells were transduced with lentivirus; 24 h after the transduction media was changed, cells were selected for stable integration using puromycin antibiotic selection at the following concentration: RWPE-1- 3 μg/mL; HEK293T, PC-3 and LNCaP: 2 μg/mL.

### R-loop slot-blot
R-loops were estimated by slot-blot as described in ref. 23. Cells were lysed in lysis buffer (100 mM Tris-HCl (pH 8.5), 5 mM EDTA, 0.2% SDS,

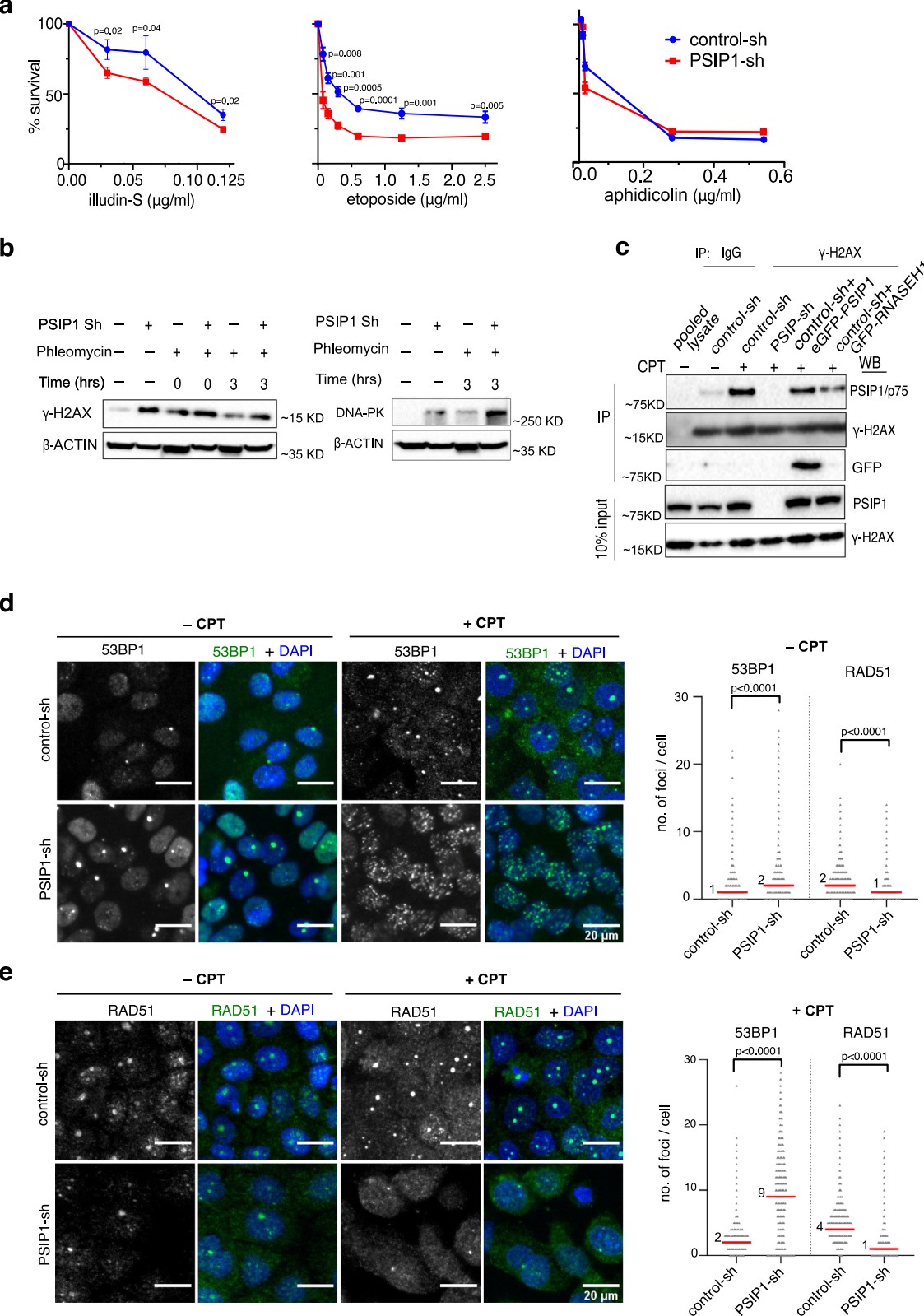

100 mM NaCl, proteinase K 0.5 mg/mL) and incubated at 55 °C overnight. Ice-cold isopropanol was added to the lysate, and the RNA-DNA was pelleted using centrifugation (12,000 RPM; 4 °C). The pellet was washed once with 70% ethanol, and the pellet was dissolved in TE buffer. The RNA-DNA was quantified using a nanodrop spectrophotometer by taking the absorption at 260 nm. Based on the quantification, different quantities (500, 250 and 125 ng) of DNA were

blotted onto the N+ nitrocellulose membrane using a slot-blot apparatus (Hoefer; slot-blot manifold). For RNASEH digestion, 2 μg of RNA-DNA was incubated with 5 units of RNASEH at 37 °C for 20 min and they were used for slotting on the membrane. The membrane was baked at 80 °C for 2 h, incubated in blocking buffer (5% skimmed milk in PBS) for 1 h, followed by incubation with S9.6 antibody (1:1000 dilution) overnight at 4 °C in a nutator. Blots were washed three times with PBST

**Fig. 4 | PSIP depletion leads to reduced HR repair at the R-loop-induced DNA damage. a** Mean % survival ± SD of PSIP-KD and control-KD RWPE-1 cells treated with different concentrations of illudin-S, etoposide and aphidicolin drugs (n = 3 independent experiments; *p*-values by multiple unpaired *t*-tests). **b** Western blot for γ-H2AX and DNA-PK (right) in control-KD and PSIP1-KD cells in cells treated with phleomycin (1 μg/mL) followed by repair for the indicated time. β-actin served as a loading control. **c** Western blots for PSIP1/p75, γ-H2AX, for γ-H2AX IPed HEK293T extracts in control, PSIP1 KD, in the presence of camptothecin (CPT@10 μM; 2 h) (+) and DMSO (−). γ-H2AX IP was also performed after overexpressing the cells with RNASEH1. IgG is a negative control; input extracts are blotted (below).

**d** Representative immunofluorescence microscopic images showing 53BP1 foci in DMSO-treated or CPT (10 μM; 2 h) treated RWPE-1 cells. The median and number of foci per cell are plotted as dot plots (*n* > 6000 cells in each group over three independent experiments; median values are indicated with a red line and *p*-values are obtained using a two-tailed Mann–Whitney test). **e** Like **d**, but for RAD51 foci. Dot plots showing the number of foci per cell with median values indicated (n > 6000 cells in each group over three independent experiments; *p*-values obtained by two-tailed Mann–Whitney test). Source data are provided as a Source Data file.

(5 min each), incubated with secondary antibody (1:1500 dilution) for 2 h at room temperature, followed by three washes in PBST 3 times (5 min each) and developed with the ECL reagents A and B (Cell signalling Technology cat no: 46935P3 and 74709P3). The same blots were stained with methylene blue to quantify total DNA.

### Determining PSIP1 binding with R-loops

To determine the binding of PSIP1 isoforms (p75 and p52) to R-loops, we used the blot binding method, a method used by Patel et al. (2020), which was followed with some modifications[56]. DNA with R-loops was extracted from HEK293T cells using the steps described in the previous section. The isolated DNA was slotted onto the membrane, blocked using 5% skimmed milk and then the membrane was incubated with either p75 protein (150 ng in 3 mL of 5% skimmed milk), or p52 protein (150 ng in 3 mL of 5% skimmed milk) or IgG protein (as control) overnight at 4 °C in a nutator. Then the membrane was washed three times with PBST solution (5 min each) and incubated with antibody (1:1000 dilution) specific to p75, p52 and IgG (incubation O/N at 4 °C in a nutator). The membranes were washed three times again using PBST and incubated with HRP coupled secondary antibody (1:1500 dilution) and the blots were developed with ECL reagents as described in the previous section.

### Immunoprecipitation

The cells were grown to ~90 % confluency, washed with PBS, treated with hypotonic solution (10 mM Tris pH 8.0, 1.5 mM MgCl2, 10 mM KCl, 0.5 mM DTT, Protease inhibitor cocktail), scrapped from the dish using a cell scrapper, and collected in the centrifuge tubes. The nuclear pellet was isolated, and the nuclear extract was prepared using NP-40 lysis buffer (50 mM Tris-HCl pH 8.0, 150 mM KCl, 1% NP-40, 1.5 mM MgCl$_2$, 0.1 mM DTT, 0.2 mM EDTA and Protease Inhibitor Cocktail). The nuclear extract was subjected to pre-clearing by incubating 500 μL of nuclear lysate with 50 μL of Protein-A Dynabeads (4 °C in a nutator for 1 h). After removal of the beads, a part of the lysate was aliquoted as the input control and the remaining lysate was incubated with the specified primary antibody (5 μg of antibody for 500 μL of nuclear lysate) overnight at 4 °C. Washed Protein-A Dynabeads were added to the lysate, incubated at 4 °C for 1 h, and pulled down using the magnet. The supernatant was discarded, and the beads were washed three times (10 min each) with NP-40 lysis buffer. The proteins bound to the beads were extracted using a Bolt Nupage 4x sample loading buffer and loaded onto the gel for the detection of different proteins by western blotting.

### Western blotting

For western blotting, cells were harvested, pelleted, and washed with PBS once and lysates were prepared using RIPA buffer (50 mM Tris, pH 8.0, 150 mM NaCl, 1% NP-40, 0.5% Sodium deoxycholate, 0.1% SDS, protease inhibitors and Benzonase; incubation on ice for 30 min with brief sonication for 5 min). Extracts were cleared by centrifugation at 13,000 RPM for 10 min at 4 °C, sample buffer, and reducing agents were added and the lysate was boiled for 5 min in a boiling water bath before cooling on ice. The sample lysates were loaded onto the True-PAGE precast gels and were subjected to electrophoresis at 95 V for

90 min. Proteins were transferred to the PVDF membrane using a Trans-blot turbo transfer system (Bio-Rad). Membranes were blocked with every blot blocking buffer (Bio-Rad), incubated with primary antibody (~1:1000 dilution in blocking buffer) overnight at 4 °C, followed by three washes with 0.05% PBST, incubated with HRP coupled secondary antibodies (1:1500 dilution in blocking buffer) for 2 h at room temperature, followed by three times wash with 0.05% PBST. Membranes were developed with ECL reagents.

### Proximity ligation assay (PLA)

Control and PSIP1-KD RWPE1 cells were seeded in 96-well plates (glass bottom). PLA was performed using the Duolink In Situ Red Starter kit (Mouse/Rabbit) according to the manufacturer protocol (Merck, DUO92101).

### Clonogenic assay

Five hundred cells for each cell type were plated in triplicates for each treatment per well in a 6-well plate. The cells were allowed to grow and develop colonies for 10–14 days after the treatment of the indicated drugs. The colonies were washed and fixed with methanol, stained with crystal violet, and counted manually to calculate the survival fraction using the following formula. Survival fraction = plating efficiency of treatment/plating efficiency of control X-100.

### Immunofluorescence

For immunofluorescence of R-loops, the exponentially growing cells were washed once with PBS and fixed using cold methanol at −20 °C for 15 min. For immunofluorescence of γ-H2AX and 53BP1, cells were fixed using 4% paraformaldehyde for 20 min. After fixation, the cells were washed with PBS, permeabilised using 0.3% Triton-X-100, blocked using 3% BSA for 1 h, and incubated with primary antibody (1:250 dilution) at 4 °C for overnight. After washing with PBST three times, the cells were incubated with secondary antibodies conjugated to different fluorophores (1:750 dilution) and were imaged in the laser-based InCell high-content imaging system (IN Cell Analyzer 6000, GE Healthcare). The nuclear levels of S9.6 or the number of γ-H2AX foci were quantified using the INCarta software attached to the imaging system. Fluorescence was quantified and plotted using GraphPad Prism 9.0 software. The same slides were also visualised in Zeiss 880 Laser Scanning Confocal Microscope and imaged.

### Plasmid overexpression

For overexpression of pEGFP-RNASEH1 (Addgene ID 108699, a kind gift from Martin Reigns, MRC-HGU, Edinburgh)[57], HA-p75-IRES-GFP, HA-p52-IRES-GFP plasmids in PSIP1 KD cells, FuGENE® HD Transfection Reagent (Promega) was used as per the manufacturer's instruction. In control cells, mock transfection was performed using IRES-GFP plasmids. The transfection efficiency was verified using the expression of GFP. Thirty-six hours after transfection, cells were used for further experiments.

### CUT&Tag

CUT&Tag was performed according to ref. 58 protocol with modifications to tissue processing as described below. Experiments were

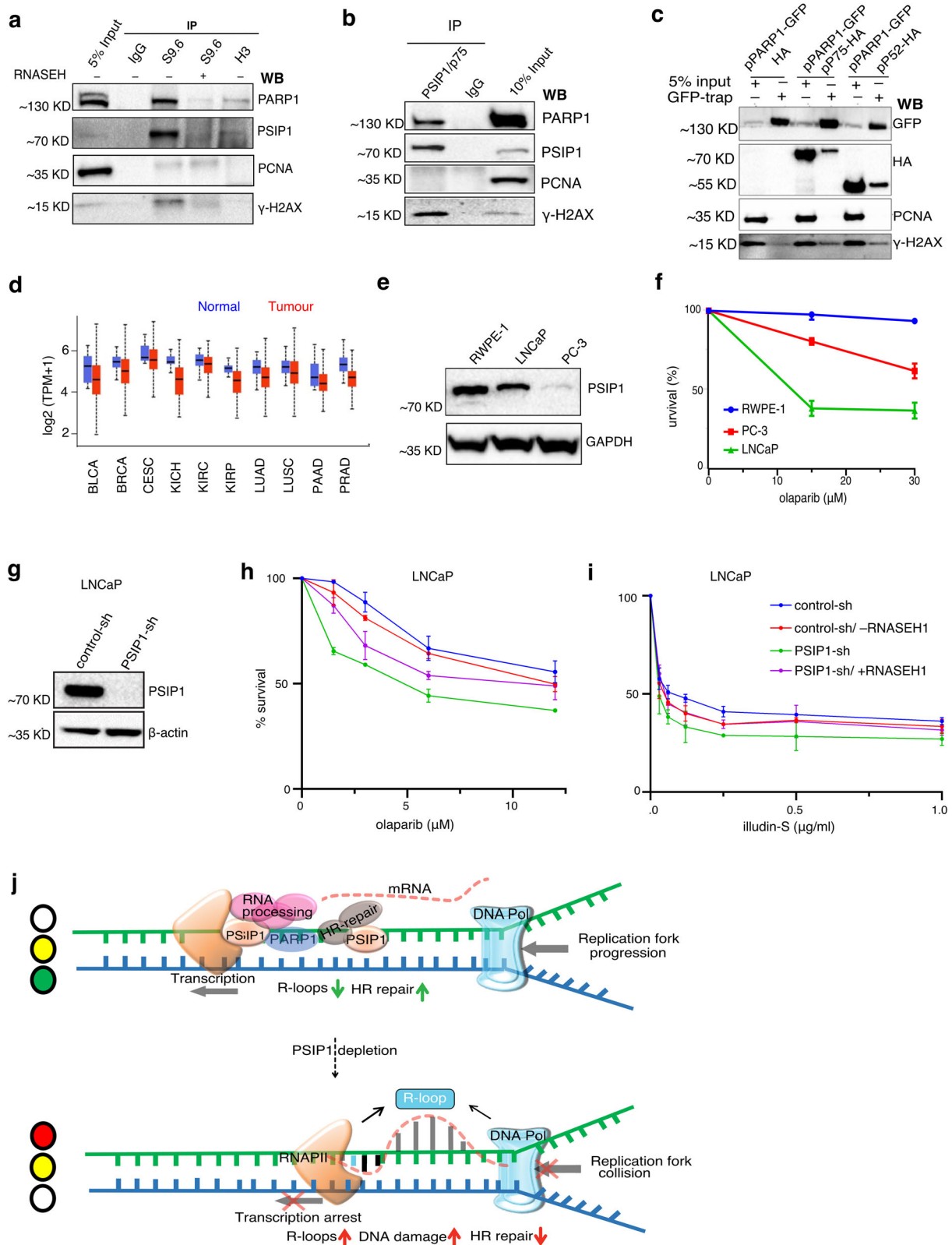

performed in biological duplicates from each cell type. ~100,000 cells were pelleted by centrifugation for 3 min at 600 x g at room temperature and resuspended in 500 µL of ice-cold NE1 buffer (20 mM HEPES-KOH pH 7.9, 10 mM KCl, 0.5 mM spermidine, 1% Triton-X-100, and 20 % glycerol and cOmplete EDTA free protease inhibitor tablet) and was let to sit for 10 min on ice. Nuclei were pelleted by centrifugation for 4 min at 1300 x g at 4 °C, resuspended

in 500 µL of wash buffer and held on ice until beads were ready. The required amount of BioMag Plus Concanavalin-A-conjugated magnetic beads (ConA beads, Polysciences, Inc) were transferred into the binding buffer (20 mM HEPES-KOH pH 7.9, 10 mM KCl, 1 mM CaCl$_2$ and 1 mM MnCl$_2$), washed once in the same buffer, each time placing them on a magnetic rack to allow the beads to separate from the buffer and resuspended in binding buffer. 10 µL of beads was

**Fig. 5 | PSIP1 deficiency increases the sensitivity of cancer cells to drugs that induce transcription-coupled DNA damage. a** Western blotting of S9.6, H3 and IgG immunoprecipitated (IP) from RWPE-1 cell nuclear extract with PSIP1/p75, PARP1, γ-H2AX and PCNA (negative control) antibodies. Lysate treated with RNA-SEH before the pulldown was used as a negative control. IgG and H3 IP were negative controls, and 5% of the nuclear extract was used as input. **b** Western blotting for PSIP1-IP with PSIP1, PARP1, γ-H2AX and PCNA antibodies, IgG served as a negative control. 10% of the nuclear extract was used as input. **c** Western blotting for eGFP-PARP1 and HA-PSIP1 co-IPs was performed using GFP-trap beads. αGFP ab was to detect eGFP-PARP1 and αHA to detect HA-PSIP1/p75 and HA-p52, along with γ-H2AX and PCNA antibodies. **d** TGCA expression data showing levels of PSIP1 transcripts in cancers of the bladder (BLCA; *n* = 408), breast (BRCA; *n* = 1097), cervical (CESC; *n* = 305), kidney chromophobe (KICH; *n* = 67), kidney renal clear cell (KIRC; *n* = 533), Kidney renal papillary cell (KIRP; *n* = 290), lung (LUAD; *n* = 515), lung squamous cell (LUSC; *n* = 503), pancreatic (PAAD; *n* = 178) and prostate (PRAD; n = 497). The data was obtained from the UALCAN online tool. The centre line indicates a median value; the boxes and whiskers indicate 25th to 75th and 10th to 90th percentiles, respectively. **e** Western blot showing the levels of PSIP1 in prostate normal and cancer cells and their sensitivity to olaparib as determined by CCK-8 assay (**f**). Mean survival ± SD is plotted (*n* = 3 independent experiments). **g** Western blot confirming the depletion of PSIP1 by shRNA in LNCaP cells. **h** Cell survival assay showing % survival of control and PSIP1 knockdown LNCaP cells with indicated doses of olaparib determined using the CCK-8 assay (mean ± SD; *n* = 3 independent experiments). **i** Similar to **h** but for indicated doses of illudin-S. **j** Working model showing the role of PSIP1 in reducing R-loop level at transcription sites to minimise transcription-replication conflict leading to DNA damage. Source data are provided as a Source Data file.

added to each tube containing cells and rotated on an end-to-end rotator for 10 min. After a pulse spin to remove liquid from the cap, tubes were placed on a magnet stand to clear, the liquid was withdrawn, and 800 µL of antibody buffer containing 1 µg of primary antibodies was added and incubated at 4 °C overnight in a nutator. Secondary antibodies (guinea pig α-rabbit antibody, Antibodies online, ABIN101961 or α-mouse antibodies) were diluted 1:100 in dig-wash buffer (5% digitonin in wash buffer), and 100 µL was added in per sample while gently vortexing to allow the solution to dislodge the beads from the sides and incubated for 60 min on a nutator. Unbound antibodies were washed in 1 mL of dig-wash buffer three times. 100 µL of (1:250 diluted) protein-A-Tn5 loaded with adapters in dig-300 buffer (20 mM HEPES pH 7.5, 300 mM NaCl, 0.5 mM spermidine with Roche cOmplete EDTA free protease inhibitor) was added to the samples, placed on nutator for 1 h and washed three times in 1 mL of dig-300 buffer to remove unbound pA-Tn5. 300 µL tagmentation buffer (Dig-300 buffer + 5 mM MgCl$_2$) was added while gentle vortexing, and samples were incubated at 37 °C for 1 hr. Tagmentation was stopped by adding 10 µL 0.5 M EDTA, 3 µL 10% SDS and 2.5 µL 20 mg/mL Proteinase K to each sample. Samples were mixed by full-speed vortexing for ~2 s and incubated for 1 h at 55 °C to digest proteins. DNA was purified by phenol: chloroform extraction using phase-lock tubes (Quanta Bio) followed by ethanol precipitation. Libraries were prepared using NEBNext HiFi 2x PCR Master mix (M0541S) with a 72 °C gap-filling step followed by 13 cycles of PCR with 10-second combined annealing and extension to enrich short DNA fragments. Libraries were sequenced in Novaseq 6000 (Novogene) with 150 bp paired-end reads. For HEK293T CUT&Tag, we followed the following protocol (https://www.protocols.io/view/cut-amp-tag-direct-for-whole-cells-with-cutac-x54v9mkmzg3e/v4).

### Transient transcriptome sequencing (TT-seq)
TT-seq was performed according to the protocol described in[59] with minor modifications. Control and PSIP1 knockdown cells were labelled with 500 µM 4-thiouridine (Sigma) for 20 min. Cells were counted and collected in TRIzol, to which 5% S2 cells labelled with 4sU for 2 h were added. RNA was chloroform-extracted, DNase-treated, and chloroform-extracted. To 60 µg total RNA was added 2× fragmentation buffer (final concentration: 75 mM Tris-Cl, pH 8.3, 112.5 mM KCl, and 4.5 mM MgCl$_2$) and heated to 95 °C for 5 min. Fragmentation was stopped by adding EDTA to 50 mM and placing samples on ice. RNA was ethanol-precipitated and resuspended in H$_2$O. Fragment sizes were checked on the Bioanalyser (peak size of ~800 nucleotides). The biotinylation reaction was performed with 0.025 mg/mL MTSEA-Biotin XX (Biotum) in reaction buffer (20% *N,N*-dimethyl-formamide, 1 mM EDTA, 10 mM HEPES pH 8) for 45 min in the dark. Labelled RNA was chloroform-extracted ethanol-precipitated and resuspended in 90 µL H$_2$O. 75 microliters of DynabeadsTM M-280 streptavidin (Thermo Fisher) were prewashed

with decon solution (0.1 M NaOH + 50 mM NaCl), 2× 100 mM NaCl, 2× high salt buffer (100 mM Tris-Cl pH 7.4, 10 mM EDTA pH 8, 1 M NaCl, 0.05% (vol/vol) Tween 20), and resuspended in high salt buffer. Labelled RNA was denatured by heating to 65 °C for 5 min and placing on ice for 2 min. Ten microliters of high salt buffer were added to labelled RNA. Prewashed beads were placed on the magnet, the supernatant discarded, and then beads were resuspended in the labelled RNA. Beads and RNA were rotated for 30 min in the dark. Beads were then washed 4× for 1 min with high salt buffer. Labelled RNA was eluted 2× with 100 mM DTT (1,4-dithiothreitol). RNA was cleaned up with an RNeasy MinElute Clean-up Kit (Qiagen Cat. No 74204). 250 ng of RNA was used for library preparation using the CORALL total RNAseq library preparation kit (Lexogen Cat. no. 147.24) with the modification of shortening the fragmentation step to 3 min and using 8 cycles of PCR amplification. Libraries were sequenced in Novaseq 6000 (Novogene) with 150 bp paired-end reads.

### CCK-8-assay
Seven thousand cells were plated on a 96-well plate for overnight attachment. The next day, cells were added with different drug concentrations and cultured for 24-48 h in the incubator. Fresh media containing 50 µg/mL of WST-8 dye was replaced and incubated for 1 h. Then the absorbance was recorded at 450 nm, and the absorbance value was used to calculate the percentage of viable cells.

### EdU incorporation assay
EdU incorporation was performed using Click-iT™ Plus EdU Alexa Fluor™ 647 Assay Kit (Invitrogen). EdU (5-ethynyl-2'-deoxyuridine, 20 mM) was added to the culture dishes with cells and grown for 4 h. Then, cells were fixed using 3.8% formaldehyde for 15 min at room temperature and washed twice with 3% BSA in PBS. Cells were permeabilised using 0.5% Triton-X-100 (20 min) and stained using the Click-iT cocktail following the vendor's protocol. Cells were imaged under IN Cell Analyzer 2200 (GE Healthcare Life Sciences), and the fluorescent intensity was quantified (from at least 5000 cells) using IN Carta Image Analysis Software (Molecular Devices).

**Cell cycle analysis by flowcytometry.** The exponentially growing cells were harvested, and stained with DAPI and cells were acquired using an ACEA Novocyte 3000 flow cytometer. The cell cycle analysis was performed using NovoExpress Software attached to the machine.

### Data analysis: CUT&Tag
The pair-end sequencing reads generated for the CUT&Tag were trimmed for the sequencing adapters with a trimmomatic tool, followed by mapping onto the human genome hg38 using *bowtie2*. The reads were aligned using following parameters: *–very-sensitive-local –no-unal –no-mixed –no-discordant –phred33 -I 10 -X 700*. The data was

processed using SAMtools to generate bam files and sort them. All the replicates were pooled using *SAMtools merge*. Merged bam files were used to generate bed files, bedgraph and bigwig.

## Peak calling

Peaks on the mapped reads were called using the *SEACR* tool using IgG as a background with *"norm and relaxed"* options. For each replicate, the peaks were called using both individual IgG replicates as background. The peaks obtained were pooled, only the peaks consistent in both replicates were retained, and coordinates were merged. These peaks were used for further analyses.

Peaks were processed using the *bedtools intersect* option to filter the peaks unique to either WT (PSIP peaks) or KD (γ-H2AX and S9.6 peaks). Peak distribution across genomic landmarks (TSS, gene body or intergenic) was done on the coordinates obtained from the UCSC genome browser using *bedtools intersect*.

## Bigwig generation and plotting

The signal files for merged samples were generated using the deep-Tools bamCoverage option with the following parameters: *−binSize 20 −normalise Using CPM −scaleFactor −smoothLength 60 −extended 150 −centerReads*. The signal was normalised between WT and KD for each target by comparing the reads mapping to the *E. coli* genome. The bigwigs generated were used for viewing on the genome browser or plotting as a heatmap or average summary plot on the genomic landmarks and peaks. Matrices generated through *computeMatrix* with *reference-point or scale-region option* were used as input for heatmap *(plotHeatmap) or* average summary *(plotProfile), or* violin *(R package ggplot2)* plots. The genome-browser views were captured on the UCSC genome browser.

## Data analysis: RNAseq and TTseq

Pair-end reads for the RNA-seq, and TT-seq were aligned against the human genome (hg38) via STAR aligner following the Bluebee-CORALL mapping pipeline. The replicates were merged for each sample using SAMtools merge, followed by filtering out the multi-mapped reads by defining MAPQ as 255. The bigwigs were generated using the deepTools bamCoverage tool with the option to normalise using RPKM. These bigwigs were used for comparing the signal in the genome browser or plotting heatmaps, average summary plots or violin plots on the genes and peaks datasets. Differential gene expression for the RNAseq was performed through the DESeq2 package. The fragments count matrix was generated using the Subreads feature counts option. This count matrix was subjected to DESeq2, and the result obtained from the analysis was used to plot as a Volcano plot.

## GC skew analysis

GC skew for the S9.6, PSIP1 and γ-H2AX CUT&Tag peaks was calculated using the ratio of $(G - C)/(G + C)$. The values were used to plot the frequency distribution. For the background frequency of the GC skew, randomly shuffled genomic coordinates were used to compare it. Further, the PSIP1, S9.6 and γ-H2AX signal distribution was compared on the GC-skewed regions of our genome[28]. The coordinates were lifted from hg18 to hg38 and used for plotting. Likewise, the signal was also plotted for the G-rich promoters[28].

## Reporting summary

Further information on research design is available in the Nature Portfolio Reporting Summary linked to this article.

## Data availability

The raw sequencing data for CUT&TAG, RNA-Seq and TT-Seq discussed in this study have been deposited in NCBI's Gene Expression Omnibus and are accessible through GEO Series accession number GSE220234. Source data are provided in this paper.

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

## Acknowledgements

We thank QMUL epigenetics hub, Madapura lab members, Roberto Bellelli (QMUL) and Urszula McClurg (University of Liverpool) for the discussions. We thank Vaclav Veverka (Charles University, Czech Republic) for sharing recombinant PSIP1 proteins, Francisco Martin-Zamora and Chema Martin (QMUL) lab for sharing purified pATn5, Martin Regins (MRC-HGU, Edinburgh) for RNASEH1 plasmid, Benjamin JE Martin (Harvard Medical School) for TT-seq protocol. We thank Blizard core facilities and Luke Gammon, Maeve McLaughlin and Gary Warnes for their help in image analysis training and FACS analysis. This research utilised Queen Mary's Apocrita HPC facility, supported by QMUL Research-IT. Marie Skłodowska-Curie Actions Individual Fellowship (896079-J.S.), Medical Research Council UKRI/MRC grant and Barts charity grant (MR/T000783/1; MGU0475-M.M.P.).

## Author contributions

Conceptualisation: M.M.P., J.S., H.T.; Methodology: M.M.P., J.S., H.T., M.P., G.N.B., F.B.; Investigation: M.M.P., J.S., M.P., H.A.; Visualisation: M.M.P., J.S., M.P.; Funding acquisition: M.M.P., J.S.; Project administration: M.M.P.; Supervision: M.M.P., H.T.; Writing—original draft: M.M.P., J.S.; Writing—review & editing: M.M.P., J.S., H.T., M.P., G.N.B.

## Competing interests

The authors declare no competing interests.
