## [Peer Review File · Nature Communications]

PSIP1/LEDGF reduces R-loops at transcription sites to maintain genome integrityREVIEWER COMMENTS

Reviewer #1 (Remarks to the Author):

In this manuscript, Jayakumar and colleagues demonstrate a role for PSIP1/LEDGF in R-loop metabolism and genome stability. Specifically, they begin by showing that PSIP1 interacts with R-loops and other factors involved in R-loop resolution. They then show that loss of PSIP1 increases R-loops genome-wide, and that PSIP1 loss increases DNA damage signaling, at least partly due to transcription-replication conflicts. In addition, PSIP1 loss increases sensitivity to select damaging agents associated with R-loop formation, and this appears to function at least partially by affecting HR, as measured by 53BP1/Rad51 foci.

This manuscript is a significant advance, as it 1) elaborates on the function of PSIP1 in R-loop biology and genome stability, and 2) builds upon established work connecting transcription factors and R-loop suppression. In my opinion it is appropriate for publication in Nature Communications if the following issues are addressed:

Major issues:

1. One of the major questions in the R-loop field is that many factors that affect transcriptional elongation or splicing of Pol II transcripts are likely involved in suppressing R-loops, and PSIP1 may be no exception. However, whether these are epistatic (i.e., function in the same pathway) with one another is unclear. It would make the paper stronger if the authors could demonstrate what happens to R-loops (globally is sufficient) in WT and PSIP KO cells with and without inhibition of the spliceosome, which can be done with pladienolide B, or loss of Aquarius (or other spliceosomal factor).
2. Related to the above: What domain(s) of PSIP1 are required for the R-loop phenotype? Specifically, are the PWWP or the TFIIS-NTD domains important for this phenotype? This can be done with simple rescue experiments of their knockdown or knockout cell lines.
3. Does loss of PSIP1 cause increased sensitivity to PARP inhibitors? This would significantly increase the impact of the paper.
4. Can the illudin-S or etoposide sensitivity be rescued by overexpressing RNaseH1?

Minor issues:

1. The PLA experiment in Figure 1 would be improved with additional controls. Specifically, either overexpression of RNase H1 or knockout of PSIP1 should abrogate the PLA foci. In its current form it is difficult to come to any conclusions without such a control.
2. It would help the reader if the authors could demonstrate (e.g., with a Venn diagram) the overlap between R-loop localization genome wide upon loss of PSIP and the localization of PSIP (Figure 1K).
3. It is unclear what the light purple sample (with no line) represents in the TT-Seq metagene plot (Figure 3D).
4. The third paragraph reads like a list of facts related to PSIP1 function. This should be rewritten. There are also some minor issues with the English throughout the manuscript; it could be improved with more careful proofreading.

Reviewer #2 (Remarks to the Author):

This manuscript studies the relationship of the PSIP1 factor with R-loops and the consequences of its dysfunction in R-loop homeostasis and genetic instability. Using standard methods of IF, immunoprecipitation with the S9.6 antibody, analysis of the impact of RNH in some cases, PLA of different elements and DNA damage as determined by gammaH2AX, plus genome-wide seq analyses authors conclude that PSIP1 is a novel factor that binds R-loops and function to reduce R-loop

accumulation and prevent genetic instability by promoting HR repair. In addition, authors show that 2 prostate cancer cell lines with low PSP1 levels display more sensitivity to PARPi, claiming thus that PSPI inactivation could enhance cancer treatment.

The manuscript thus reports yet another protein that when depleted causes an enhancement of R loops and the expected increase in gammaH2AX likely caused by transcription-replication conflicts, even though this is not demonstrated along the manuscript. The analysis of the effect of PSIP1, previously identified as an R-loop-interacting protein in global approaches, is one more of the increasing list of proteins affecting R-loop. Unfortunately, the study does not convincingly show a role for PSIP1 in R-loop control and does not provide a mechanism by which to explain how this factor whose function is related to chromatin and transcription, would control R-loop levels and how would relate to its function. It is highly relevant that authors do not confirm their conclusions by reverting the many phenotypes they are proposing to be mediated by the high levels of R-loops generated in PSIP1 depleted cells by RNH.

Specific comments:

- The argument of why to focus on PSIP1 is not too strong. Just based on proteomic analysis of others, giving as a fact that PSIP1 is an R-loop factor. It would be necessary to demonstrate that PSIP1 binds R-loops by in vitro studies. In this sense, the studies in cells does not directly answer this question. A condition arresting the RNA polymerase might be associated with a concomitant accumulation of DNA-RNA hybrids that can explain their association with PSIP1 in IP experiments, given that it may be a protein engaged in transcription together with others.

- In S9.6-PSIP PLA experiment for Fig. 1E & S1B and others PLA experiments within the manuscript single antibodies controls should be included to ensure the specificity of the signals. Additionally, the suppression of S9.6-PSIP interaction upon RNaseH1 overexpression would ensure that S9.6-PSIP signal is mediated by DNA-RNA hybrids and no other possible RNA species detected by the S9.6 antibody. Alternatively, interaction of PSIP to R-loop-enriched regions and its dependency on R-loop could be assessed by PSIP ChIP at specific R-loop-enriched regions in the absence or presence of RNaseH1.

- Regarding Fig. 1F-H, S1D & S2B-D. Authors show accumulation of S9.6 signal upon PSIP depletion in MEF, RWPE-1 and HEK293T cells using slot blot for genomic DNA and hybridization with the S9.6 antibody. However, the specificity of the signal assessed by the treatment of the samples with RNase H is only showed for PSIP1 knockout MEFs. The in vivo overexpression of RNaseH1 and the quantifications of all the experiments is required to ensure the specificity of the signals. In addition, quantifications for experiments in figures S2B and D and the normalization of S9.6 intensity for RNase H treated samples in figure S1D should be included.

- Additionally, a more direct detection of DNA-RNA hybrids by DRIP-qPCR at some coincident PSIP1- and gH2AX-enriched genomic regions, as determined by genome-wide analyses, in control and PSIP1-depleted conditions (treated and not treated with RNaseH) are required. In addition, analysis of the transcription levels of the selected targets would ensure that differences in DNA-RNA hybrid levels in PSIP1-depleted conditions are not due to changes in gene expression.

- Related to Fig. 1I and in agreement with the previous comments, the specificity of S9.6 immunofluorescence signal should be controlled in RNaseH1 overexpressing cells. S9.6 foci quantified in Figure S1F are difficult to see in Figure 1I. A better-quality picture of S9.6 immunofluorescence as well as a more detailed explanation regarding R-loop foci quantification and whether nucleolar signal was excluded for the analysis would help readers to understand the analysis.

- In lines 140-141 authors indicate that RNAseq data analysis showed minimal changes in the expression of 186 genes that are known to be involved in R-loop homeostasis (Fig. S2E). Which genes are included for the analysis?

- In lines 142-143 authors say: "All this evidence shows that the elevated R-loop level is due to the

direct effect of PSIP1 depletion but not due to the indirect consequence of the altered transcriptional programme". Considering that RNA-seq data analysis show that several genes involved in R-loop homeostasis are downregulated in PSIP1 knockdown conditions, it would be important to determine in a more quantitatively way the coincidence between PSIP1 peaks and S9.6 peaks observed in PSIP1-sh conditions, for example, using Venn diagrams.

- In S9.6 CUT & Tag experiment, a control with RNase H treatment is missing

- Authors show that PSIP1 depletion leads to increased DNA damage by detecting γ H2AX signal/foci and γ H2AX CUT&TAG in PSIP1 depleted cells in Fig.2. However, the fact that it depends on R-loops should be further demonstrated experimentally. This is the case of the need to suppress γ H2AX foci accumulation by RNase H1 overexpression in PSIP1 depleted cells in Fig. 2A and C.

- In Fig. 2H error bars and statistical analysis are missing. Also, it is not clear whether relative western blots signals are normalized versus the loading control and how many repetitions are included for the analysis. RNaseH expression and triptolide treatment in Fig. S3D leads to an increase in γ H2AX signal compared to control cells and therefore the suppression effect is hard to see at least data are normalized versus the cont-sh condition, which is not very convincing.

- In Fig. 2E authors say that γ H2AX peaks gained in PSIP1 cells coincide with PSIP1 enriched regions. Additional quantitative analysis would help demonstrate this result. How many PSIP1 peaks coincide with γ H2AX and S9.6 peaks upon PSIP1 depletion? How many γ H2AX peaks gained in PSIP1 cells coincide with PSIP1 peaks?

- In Fig. 3A-B metaplots of average profiles of PSIP1, R-loops and γ H2AX signals across gene bodies and promoters in control and PSIP1-sh cells should be included to complement the visualization of the distributions of the signals in Fig. 3A.

- In lines 195-202 and Fig. 3D and E authors investigate whether R-loops induced upon PSIP1 depletion leads to transcriptional arrest. TT-seq data show a reduction in the nascent transcript levels at R-loop peaks upon PSIP1 depletion but it might imply a global reduction of nascent transcript levels nor necessarily related to R-loops. In order to confirm the specificity to R-loop regions, a comparison between R-loop-enriched regions and R-loop-free regions should be performed. Furthermore, authors say that "triptolide mediated transcription inhibition reversed the elevated R-loop levels in PSIP1-KD (Fig. 3E). This suggests the role of PSIP1 in reducing the R-loop burden during ongoing transcription, accumulation of these R-loops can otherwise lead to transcriptional arrest". However, the fact that transcription inhibition leads to R-loop reduction might imply that there is less RNA prone to R-loop formation but not necessarily that those R loops are responsible for transcriptional arrest.

- In Fig. 3F, S4B and lines 203-212 authors conclude that PSIP1 prevents R-loop-mediated transcription collisions by observing an increased number of collisions by RNPII-PCNA PLA and a reduction of EdU-incorporating cells. However, again, showing a dependency of those phenotypes on R-loops is required. To conclude that those collisions are R-loop dependent a suppression of the phenotypes by RNaseH expression should be shown. In addition, a reduced number of EdU-incorporating cells could also be explained by differences in the abundance of cells in different cell cycle stages (reduced number of cells in S phase in PSIP1-sh) but not necessarily slower replication rates. A more direct measurement of replication rate could be assessed by determining EdU incorporation track lengths by DNA fiber analysis or following EdU incorporation combined with DNA staining by FACS after cell cycle synchronization and releasing into the S phase.

- The enrichment of R-loops at CpG promoters (this was something already shown in the past as a general feature by F. Chedin's lab) may be a consequence of the known association of R-loops with high GC-skew, but without a functional analysis of the physiological meaning at CpG promoters, it is

unclear the relevance of this data. In addition, if PSP1 controls R-loops at CpG island promoters, how to explain hybrids at the promoters themselves. Are they transcribed then? Antisense?

- Authors conclude that PSIP1 promotes the repair of DNA damage induced at sites of transcription based on the observation of a mayor sensitivity of PSIP1 depleted cells to illudin-S, etoposide and Phleomycin compared to aphidicolin (Fig. 3A & S5B). It is unclear how it is possible to conclude this just from assays of sensitivities.

- Colocalization of γ H2AX and PSIP1 is not evident from Fig. S5 given that PSIP1 staining seems to be pan-nuclear whereas discrete γ H2AX foci are shown.

- Could the differences in 53BP1 and RAD51 foci formation in PSIP1-sh cells induced by CPT or phleomycin be explained by a differential cell cycle distribution pattern caused by PSIP1 depletion?

- In Fig. 5B and C authors relate PSIP1 levels in 2 different prostate cancer cell lines with their PARPi sensitivity. However, results show that the cell line with higher PSIP1 levels (LNCaP) is more sensitive to Olaparib than the one with lower levels of PSIP1 (PC-3), suggesting that there is not a direct correlation between those 2 facts (Olaparib sensitivity and PSIP1 levels). If Olaparib sensitivity was caused by reduced levels of PSIP1, PSIP1 overexpression in those cells lines is expected to suppress PARPi sensitivity. Besides, to conclude a role mediated by R-loops, RNH expression should partially revert the phenotype. Without these additional data, the higher PARPi sensitivity may not be related at all with R-loops.

Minor points:

- In general, numbers of replicates for immunofluorescence experiments should be stated in figures legends.

- In line 277, Fig. 5G and H should be Fig. 5D and E.

- In lines 126-127, authors referred to the generation of HEK293T and RWPE-1 cells depleted of PSIP1 but Fig S1G and S1I are missing. In legend of fig. 1G the cell line used for shRNA PSIP1 depletion is omitted.

- In Fig. 2 H & S3D details about the treatment of control cells (are transfected with an empty plasmid in order to be comparable to eGFP-RNaseH transfected cells?) and timing for RNaseH expression are missing.

- In the M&M some details are missing and should be carefully revised. For example, the compositions of some buffers are not detailed (lysis buffer, methylene blue, NP-40 lysis buffer) or the concentration of some reagents (dilution of antibodies for blots, IFs and PLAs, protein-A Dyna beads). In addition, in the clonogenic assay protocol, it might be an error in the number of plated cells or the plate used (1000 cells in a well of a 96-well plate).

Reviewer #3 (Remarks to the Author):

PSIP1/LEDGF reduces R-loops at transcription sites to maintain genome integrity.

The authors identify PSIP1 as an R-loop interacting protein and show that depletion of PSIP1 results in R-loop increases, which is correlated with increased DNA damage. Increased R-loops are also correlated with transcription arrest and TR conflicts which lead to DNA damage. PSIP1 is also suggesting to have a role in HR repair pathway, although the exact mechanism appears unclear. Finally, decreasing PSIP1 levels sensitizes cancer cells to PARP inhibitors. Throughout the manuscript the authors suggest that PSIP1 has a direct role in regulating R-loops. There is no evidence for this and the experiments performed cannot distinguish between direct vs indirect effects. Most of the experiments are observational and do not give a clear result in terms of how PSIP1 functions. Overall, while some of the observations made are interesting, many are expected, and the new results appear a little under developed for publication. Further experiments to identify a clear mechanism of action of PSIP1 in R-loop regulation will help in making a case for publication of this manuscript.

Specific Comments:

1. Figure 1C – what is a universal nuclease? There is no band in the PSIP1 input lane. Better quality western is needed.
2. Figure 1E – no negative controls are shown in PLA experiments. Also it is not clear how these dots are quantified when individual dots are not even visible.
3. Lines 96-98 – Just because R-loop levels are restored to normal when PSIP1 is reintroduced into cells do not mean this protein has a direct role in reducing R-loop levels. This is an overstatement without any supporting evidence.
4. The Chedin group has shown that the S9.6 antibody has significant cross reactivity with dsRNA in immunofluorescence experiments (PMID: 33830170). Therefore, the use of this antibody without the appropriate controls in IF experiments is not scientifically sound (Figure 1I).
5. Lines 142-143 - It is unclear how knock down experiments can distinguish between direct and indirect effects. While R-loops can increase upon PSIP1 KD, this can happen through effects on transcription elongation, splicing etc. In addition, there is no study that has comprehensively identified all genes involved in R-loop homeostasis. Where do the authors find the 186 genes that they report in Fig S2E. And just because there is no change in these genes does not mean that PSIP1 is the only factor that contributes to aberrant R-loop accumulation in these experiments. This is a major overstatement.
6. Fig 2E and 2F – PSIP1 levels in control cells at sites of h2AX accumulation look barely above background. These experiments do not look conclusive and must be repeated.
7. Figure 2G – This figure has the same quantification issue as Figure 1E.
8. Figure 3 – None of the results presented in this figure are surprising i.e it is well established that R-loops are enriched at regions with high GC skew. It is also known that R-loop accumulation can cause transcriptional arrest and TR conflict. There is no reason to believe that R-loops that accumulate because of PSIP1 depletion would be different. However, if they were different, it would be interesting to report. The data in this figure belongs in the supplement.
9. Figure 4 – The authors show that PSIP1 KD cells treated with illudin and etoposide show sustained DNA damage and conclude that PSIP1 has a role in DNA repair. What is the mechanism for this? How exactly does PSIP1 function in DNA repair?
10. Figure 4C – PSIP is pulled down with gH2AX and this is increased with CPT treatment that increases R-loops. This increase should be shown because the timing of CPT treatment has differential effects on R-loops. This result is a little confusing because R-loops are supposed to be nucleosome depleted regions. So does this imply that PSIP1 binds in the vicinity of R-loops and through gH2AX?
11. Figure 5 – If PSIP1 levels are what sensitizes cancer cells to PARP inhibitors, does PSIP1 KD in RWPE1 cells have the same effect?

Point-by-point response to Reviewer Comments

The authors thank the reviewers for their comments and feedback for improving the manuscript.

Reviewer #1 (Remarks to the Author):

In this manuscript, Jayakumar and colleagues demonstrate a role for PSIP1/LEDGF in R-loop metabolism and genome stability. Specifically, they begin by showing that PSIP1 interacts with R-loops and other factors involved in R-loop resolution. They then show that loss of PSIP1 increases R-loops genome-wide, and that PSIP1 loss increases DNA damage signaling, at least partly due to transcription-replication conflicts. In addition, PSIP1 loss increases sensitivity to select damaging agents associated with R-loop formation, and this appears to function at least partially by affecting HR, as measured by 53BP1/Rad51 foci.

This manuscript is a significant advance, as it 1) elaborates on the function of PSIP1 in R-loop biology and genome stability, and 2) builds upon established work connecting transcription factors and R-loop suppression. In my opinion it is appropriate for publication in Nature Communications if the following issues are addressed:

Response: We thanks the reviewer for reviewing our manuscript and providing with the comments to improve the manuscript. We are also glad that the reviewer found our work appropriate for publication in Nature Communications.

Major issues:

1. One of the major questions in the R-loop field is that many factors that affect transcriptional elongation or splicing of Pol II transcripts are likely involved in suppressing R-loops, and PSIP1 may be no exception. However, whether these are epistatic (i.e., function in the same pathway) with one another is unclear. It would make the paper stronger if the authors could demonstrate what happens to R-loops (globally is sufficient) in WT and PSIP KO cells with and without inhibition of the spliceosome, which can be done with pladienolide B, or loss of Aquarius (or other spliceosomal factor).

Response: As suggested by the reviewer, we carried out additional experiment to understand the interaction between PSIP1 and splicing pathway. We inhibited the splicing by treating both control and PSIP1 KD cells with pladienolide-B and estimated the R-loops by slot blot. In the control cells treated with pladienolide-B there was marginal increase in R-loop levels, however treating PSIP1 KD cells with pladienolide-B resulted in no additional increase in R-loops was seen, suggesting that the PSIP1 could be acting through the splicing pathway in reducing the R-loop burden (Below figure). We have decided not to include this data in the manuscript as we believe it needs further experimentation to support these findings. For example, a demonstration that treatment with pladienolide-B in this experiment has indeed resulted in splicing inhibition.

2. Related to the above: What domain(s) of PSIP1 are required for the R-loop phenotype? Specifically, are the PWWP or the TFIS-NTD domains important for this

phenotype? This can be done with simple rescue experiments of their knockdown or knockout cell lines.

Response: As suggested by the reviewer we performed following experiments. PSIP1 KD cells were transfected with P75-HA tagged overexpression plasmid and P52-HA tagged overexpression plasmids. P52 lacks TFIIIS-NTD domain. When we studied the R-loops accumulation in these cells it was found that both the isoforms are able to rescue the cells from R-loops accumulation mediated by PSIP1 KD, implying that the N terminal domain of PSIP1 (common to both isoforms) and not the TFIIIS-NTD domain is important for the R-loop resolution activity of PSIP1 (Fig. S2B)

3. Does loss of PSIP1 cause increased sensitivity to PARP inhibitors?

Response: We thank the reviewer for the suggestion and we carried out additional experiments in the LNCaP prostate cancer cell line to evaluate the role of PSIP1 in PARP1 sensitivity. When PSIP1 was knocked down, it increased the sensitivity of the Olaparib (PARP1 inhibitor). Excitingly, this sensitivity could be reversed by overexpression of RNASEH1 (Fig. 5H). These results imply the involvement of PSIP1 deficiency in PARPi sensitivity and R-loops' role in mediating this sensitivity.

4. Can the illudin-S or etoposide sensitivity be rescued by overexpressing RNaseH1?

Response: To address this question by the reviewer, we knocked down PSIP1 in LNCaP cells and studied the illudin-S sensitivity. Similar to olaparib, PSIP1 KD cells showed increased sensitivity to illudin-S and that could be reversed by overexpression of RNASEH1 (Fig. 5i), again indicating the PSIP1-R-loop axis in mediating the illudin-S drug sensitivity. We have included this data in the revised manuscript.

Minor issues:

1. The PLA experiment in Figure 1 would be improved with additional controls. Specifically, either overexpression of RNase H1 or knockout of PSIP1 should abrogate the PLA foci. In its current form it is difficult to come to any conclusions without such a control.

Response: We performed PLA experiment with the additional controls (Fig 1e). We show that in wild-type cells, RNASEH1 overexpression reduced the PLA foci between PSIP1 and S9.6. We have also included single antibody controls in all PLA experiments in the revised manuscript.

2. It would help the reader if the authors could demonstrate (e.g., with a Venn diagram) the overlap between R-loop localization genome wide upon loss of PSIP and the localization of PSIP (Figure 1K).

Response: As suggested by the reviewer, an upset plot was constructed using the number of unique and overlapping peaks of PSIP1, S9.6 and γ -H2AX in control and PSIP1KD cells. Upon depletion of PSIP1, 909 unique S9.6 peaks emerged that were overlapping with the PSIP1 binding in wild-type cells (Fig. 3c).

3. It is unclear what the light purple sample (with no line) represents in the TT-Seq metagene plot (Figure 3D).

Response: It is the standard deviation of the reads from PSIP1-KD cells. We have now described in the legend.

4. The third paragraph reads like a list of facts related to PSIP1 function. This should be rewritten. There are also some minor issues with the English throughout the manuscript; it could be improved with more careful proofreading.

Response: We have modified the introduction and the manuscript has also been carefully proofread.

Reviewer #2 (Remarks to the Author):

This manuscript studies the relationship of the PSIP1 factor with R-loops and the consequences of its dysfunction in R-loop homeostasis and genetic instability. Using standard methods of IF, immunoprecipitation with the S9.6 antibody, analysis of the impact of RNH in some cases, PLA of different elements and DNA damage as determined by gammaH2AX, plus genome-wide seq analyses authors conclude that PSIP1 is a novel factor that binds R-loops and function to reduce R-loop accumulation and prevent genetic instability by promoting HR repair. In addition, authors show that 2 prostate cancer cell lines with low PSP1 levels display more sensitivity to PARPi, claiming thus that PSPI inactivation could enhance cancer treatment.

The manuscript thus reports yet another protein that when depleted causes an enhancement of R loops and the expected increase in gammaH2AX likely caused by transcription-replication conflicts, even though this is not demonstrated along the manuscript. The analysis of the effect of PSIP1, previously identified as an R-loop-interacting protein in global approaches, is one more of the increasing list of proteins affecting R-loop. Unfortunately, the study does not convincingly show a role for PSIP1 in R-loop control and does not provide a mechanism by which to explain how this factor whose function is related to chromatin and transcription, would control R-loop levels and how would relate to its function. It is highly relevant that authors do not confirm their conclusions by reverting the many phenotypes they are proposing to be mediated by the high levels of R-loops generated in PSIP1 depleted cells by RNH.

Response: The authors thank the reviewer for reviewing the manuscript and offering valuable suggestions that really helped in improving the manuscript. We have carried out additional experiments addressing the deficiency pointed out by the reviewer. As indicated by the reviewer, we have carried out many rescue experiments in PSIP1-depleted cells. We have included followed figures in the revised manuscript.

- R-loop slot blot (Fig. 1i)
- R-loop immunofluorescence (Fig. 1h)
- Overexpression of P75 and P52 isoforms in PSIP1-depleted cells also resulted in the reversal of R-loop accumulation caused by PSIP1 depletion. (Fig. S2b)
- Overexpression of RNASEH1 resulted in the reversal of DNA damage in PSIP1-depleted cells, (reduced γ -H2AX and 53BP1 foci).
- Reversal of γ -H2AX foci (Fig. 2b)
- 53BP1 foci (Fig. 2c)
- 53BP1 western blotting (Fig. 2d)

- RNASEH1 overexpression also resulted in the reversal of the number of R-loop peaks and γ -H2AX peaks seen in the PSIP1-depleted cells by CUT&Tag (Fig. 2e and f).
- RNASEH1 overexpression also resulted in the reversal of the drug sensitivity mediated by the depletion of PSIP1 against olaparib and illudin-S in prostate cancer cells (Fig. 5h and i).

Specific comments:

- The argument of why to focus on PSIP1 is not too strong. Just based on proteomic analysis of others, giving as a fact that PSIP1 is an R-loop factor. It would be necessary to demonstrate that PSIP1 binds R-loops by in vitro studies. In this sense, the studies in cells does not directly answer this question. A condition arresting the RNA polymerase might be associated with a concomitant accumulation of DNA-RNA hybrids that can explain their association with PSIP1 in IP experiments, given that it may be a protein engaged in transcription together with others.

Response: As per the reviewer's suggestion, we experimented to see the binding ability of PSIP1 to the R-loops in vitro. For this, we extracted R-loops from the cells, and blotted on to the nitrocellulose membrane, followed by incubation with PSIP1 protein. After incubation with recombinant p52 and p75 proteins. Both the p75 and p52 isoforms bound to the R-loops. We have included this data in the revised manuscript (Fig. 1C)

- In S9.6-PSIP PLA experiment for Fig. 1E & S1B and others PLA experiments within the manuscript single antibodies controls should be included to ensure the specificity of the signals. Additionally, the suppression of S9.6-PSIP interaction upon RNaseH1 overexpression would ensure that S9.6-PSIP signal is mediated by DNA-RNA hybrids and no other possible RNA species detected by the S9.6 antibody. Alternatively, interaction of PSIP to R-loop-enriched regions and its dependency on R-loop could be assessed by PSIP ChIP at specific R-loop-enriched regions in the absence or presence of RNaseH1.

Response: As suggested by the reviewer, the number of foci observed in single antibody is included in the quantification. We have also included new data from RNaseH1 overexpressed cells (Fig 1e, Fig 1h, 2b, 5h and 5i).

- PLA data for S9.6 and PSIP1 with single antibody controls and reversal by RNASEH1 overexpression (Fig. 1e)
- PLA data for S9.6 and γ -H2AX with single antibody controls (Fig. 2g)
- PLA data for PCNA and RNAPII with single antibody controls and reversal by RNASEH overexpression (Fig. 3f)
- R-loop peaks enriched after PSIP1 depletion and their reversal by RNASEH overexpression by CUT&Tag (Fig. 2e)

- Regarding Fig. 1F-H, S1D & S2B-D. Authors show accumulation of S9.6 signal upon PSIP depletion in MEF, RWPE-1 and HEK293T cells using slot blot for genomic DNA and hybridization with the S9.6 antibody. However, the specificity of the signal assessed by the treatment of the samples with RNase H is only showed for PSIP1 knockout MEFs. The in vivo overexpression of RNaseH1 and the quantifications of all

the experiments is required to ensure the specificity of the signals. In addition, quantifications for experiments in figures S2B and D and the normalization of S9.6 intensity for RNase H treated samples in figure S1D should be included.

Response: To verify the specificity of the S9.6 signal, as suggested by the reviewer, we have also included the RNASEH control data for RWPE-1 cells (Fig. S1f) and HEK293T cells (Fig. S1i).

We have also carried out additional experiments by overexpressing RNASEH1 in the cells. After RNASEH1 overexpression, there was a reversal in the R-loop accumulation in the PSIP1-depleted cells. This was seen in the R-loop slot blot (Fig. 1i), R-loop immunofluorescence (Fig. 1h) and R-loop CUT&Tag experiment (Fig. 2 e & 2f).

- Additionally, a more direct detection of DNA-RNA hybrids by DRIP-qPCR at some coincident PSIP1- and gH2AX-enriched genomic regions, as determined by genome-wide analyses, in control and PSIP1-depleted conditions (treated and not treated with RNaseH) are required. In addition, analysis of the transcription levels of the selected targets would ensure that differences in DNA-RNA hybrid levels in PSIP1-depleted conditions are not due to changes in gene expression.

Response: Since R-loops detection using CUT &Tag includes a PCR amplification step, repeating PCR for qPCR is not ideal. However, further additional experiments after the overexpression of RNASEH1 show that the R-loop peaks observed upon PSIP1 depletion were reversed/reduced after the RNASEH1 overexpression (Fig. 2 e & 2f).

We studied the nascent RNA synthesis in R-loop accumulation sites and found them to be less than the control (Fig. 3d). Since the transcription at these sites are less, they may not be the reason for the R-loop accumulation. On the contrary, this reduced RNA synthesis could be due to the R-loop accumulation mediated by PSIP1 depletion. In the other places where R-loop accumulation is not there, we did not see any significant reduction in the transcription upon PSIP1 depletion.

- Related to Fig. 1I and in agreement with the previous comments, the specificity of S9.6 immunofluorescence signal should be controlled in RNaseH1 overexpressing cells. S9.6 foci quantified in Figure S1F are difficult to see in Figure 1I. A better-quality picture of S9.6 immunofluorescence as well as a more detailed explanation regarding R-loop foci quantification and whether nucleolar signal was excluded for the analysis would help readers to understand the analysis.

Response: We have performed additional experiments by overexpressing RNASEH1. The PSIP1 depletion led to the increase in R-loop accumulation and the overexpression of RNASEH1 led to the reversal of this accumulation. In addition to the use of high-content microscopy for quantification, we have now performed confocal microscope imaging to improve the quality of the images (Fig. 1h and Fig S2a).

For further quantification of R-loop fluorescence, the images were grabbed using InCell high-content imaging system and analysed using InCarta software attached to the imaging system. In the InCarta software the R-loop fluorescence coming from the entire nuclear region is calculated. We also developed an analysis pipeline using

developer software where the fluorescence coming from the nucleolar region was excluded and the non-nucleolar nuclear fluorescence was calculated. When we compared the results obtained from these two pipelines, both analyses were giving the same trend. Hence we have used a straightforward analysis pipeline using InCarta software, where in the r-loop fluorescence from nuclear region was calculated and plotted. For this analysis fluorescence from 3000-5000 cells were analysed.

- In lines 140-141 authors indicate that RNAseq data analysis showed minimal changes in the expression of 186 genes that are known to be involved in R-loop homeostasis (Fig. S2E). Which genes are included for the analysis?

Response: The list of genes reported to have a role in R-loop homeostasis was pooled from the literature and used for this analysis. We are including this list of genes with the relevant references in the revised supplementary section of the manuscript (Supplementary Table 1).

In lines 142-143 authors say: "All this evidence shows that the elevated R-loop level is due to the direct effect of PSIP1 depletion but not due to the indirect consequence of the altered transcriptional programme". Considering that RNA-seq data analysis show that several genes involved in R-loop homeostasis are downregulated in PSIP1 knockdown conditions, it would be important to determine in a more quantitative way the coincidence between PSIP1 peaks and S9.6 peaks observed in PSIP1-sh conditions, for example, using Venn diagrams.

Response: As suggested by the reviewer, an upset plot was constructed using the number of unique and overlapping peaks of PSIP1, S9.6 and γ -H2AX in control and PSIP1KD cells. Upon depletion of PSIP1, 909 unique S9.6 peaks emerged that were overlapping with the PSIP1 binding in wild-type cells (Fig. 3c).

- In S9.6 CUT & Tag experiment, a control with RNase H treatment is missing

Response: As suggested by the reviewer, we have carried out additional experiments by overexpressing RNASEH1, followed by CUT&Tag. The RNASEH1 overexpression resulted in the reversal of S9.6 peak accumulation mediated by PSIP1 depletion. We have included this data in the revised manuscript (Fig 2e and f).

- Authors show that PSIP1 depletion leads to increased DNA damage by detecting \square H2AX signal/foci and \square H2AX CUT&TAG in PSIP1 depleted cells in Fig.2. However, the fact that it depends on R-loops should be further demonstrated experimentally. This is the case of the need to suppress \square H2AX foci accumulation by RNase H1 overexpression in PSIP1 depleted cells in Fig. 2A and C.

Response: As suggested by the reviewer, we have estimated the \square H2AX level and foci after overexpressing RNASEH1 in the cells. The RNASEH1 overexpression reduced the extent of \square H2AX foci accumulated upon PSIP1 depletion. This data is included in the revised manuscript (Fig. 2b, 2e, and 2f).

Reversal of S9.6 and \square H2AX peaks by RNASEH overexpression in CUT&Tag

- In Fig. 2H error bars and statistical analysis are missing. Also, it is not clear whether relative western blots signals are normalized versus the loading control and how many repetitions are included for the analysis. RNaseH expression and triptolide treatment in Fig. S3D leads to an increase in γ H2AX signal compared to control cells and therefore the suppression effect is hard to see at least data are normalized versus the cont-sh condition, which is not very convincing.

Response: We have repeated this experiment and now we have included this data separately in the manuscript by incorporating the suggestion given by the reviewer (Fig. 2b-f). Hence we have removed the old data from the manuscript.

- In Fig. 2E authors say that γ H2AX peaks gained in PSIP1 cells coincide with PSIP1 enriched regions. Additional quantitative analysis would help demonstrate this result. How many PSIP1 peaks coincide with γ H2AX and S9.6 peaks upon PSIP1 depletion? How many γ H2AX peaks gained in PSIP1 cells coincide with PSIP1 peaks?

Response: As suggested by the reviewer, additional quantitative analysis was performed. We saw a gain of 4560 peaks, out of which 558 peaks coincide with PSIP1 peaks. Further, 252 peaks of γ H2AX and S9.6 overlap, out of which 239 peaks overlap with PSIP1 peaks. By integrating all these data, we have created an upset plot that has now been included in the revised manuscript (Fig.3c).

- In Fig. 3A-B metaplots of average profiles of PSIP1, R-loops and γ H2AX signals across gene bodies and promoters in control and PSIP1-sh cells should be included to complement the visualization of the distributions of the signals in Fig. 3A.

Response: New heatmaps in Figure 3b show enrichment of PSIP/p75 in control and S9.6 and H2AX γ in PSIP knockdown at promoters and gene bodies.

- In lines 195-202 and Fig. 3D and E authors investigate whether R-loops induced upon PSIP1 depletion leads to transcriptional arrest. TT-seq data show a reduction in the nascent transcript levels at R-loop peaks upon PSIP1 depletion but it might imply a global reduction of nascent transcript levels nor necessarily related to R-loops. In order to confirm the specificity to R-loop regions, a comparison between R-loop-enriched regions and R-loop-free regions should be performed. Furthermore, authors say that "triptolide mediated transcription inhibition reversed the elevated R-loop levels in PSIP1-KD (Fig. 3E). This suggests the role of PSIP1 in reducing the R-loop burden during ongoing transcription, accumulation of these R-loops can otherwise lead to transcriptional arrest". However, the fact that transcription inhibition leads to R-loop reduction might imply that there is less RNA prone to R-loop formation but not necessarily that those R loops are responsible for transcriptional arrest.

Response: We have included a new TTseq analysis in the revised manuscript. Although we see a global reduction in nascent transcription, we see a strong reduction in TTseq signal at promoter regions where PSIP1 and R-loops are enriched (green dotted box in Fig 3d).

We agree with the reviewer regarding the second point. However, our aim was to investigate whether inhibiting transcription could reduce the γ H2AX levels that accumulate due to R-loops. We have amended this text in the revised manuscript.

- In Fig. 3F, S4B and lines 203-212 authors conclude that PSIP1 prevents R-loop-mediated transcription collisions by observing an increased number of collisions by RNAPII-PCNA PLA and a reduction of EdU-incorporating cells. However, again, showing a dependency of those phenotypes on R-loops is required. To conclude that those collisions are R-loop dependent a suppression of the phenotypes by RNaseH expression should be shown. In addition, a reduced number of EdU-incorporating cells could also be explained by differences in the abundance of cells in different cell cycle stages (reduced number of cells in S phase in PSIP1-sh) but not necessarily slower replication rates. A more direct measurement of replication rate could be assessed by determining EdU incorporation track lengths by DNA fiber analysis or following EdU incorporation combined with DNA staining by FACS after cell cycle synchronization and releasing into the S phase.

Response: As suggested by the reviewer, we have performed RNAP-II-PCNA PLA after RNASEH1 overexpression and the results are included in the revised manuscript (Fig. 3f). We find that the overexpression of RNASEH1 reversed the association between the RNAPII and PCNA indicating the reduced transcription-replication conflicts (Fig. 3f).

We also find that upon PSIP KD there is no significant change in the cell cycle phases of the cells. The cell cycle data has been included in the revised manuscript (Fig S5b and c).

- The enrichment of R-loops at CpG promoters (this was something already shown in the past as a general feature by F. Chedin's lab) may be a consequence of the known association of R-loops with high GC-skew, but without a functional analysis of the physiological meaning at CpG promoters, it is unclear the relevance of this data. In addition, if PSP1 controls R-loops at CpG island promoters, how to explain hybrids at the promoters themselves. Are they transcribed then? Antisense?

Response: As the reviewer has pointed that R-loops at CpG promoters has been shown as a general feature, we have now moved this data to the supplementary.

- Authors conclude that PSIP1 promotes the repair of DNA damage induced at sites of transcription based on the observation of a mayor sensitivity of PSIP1 depleted cells to illudin-S, etoposide and Phleomycin compared to aphidicolin (Fig. 3A & S5B). It is unclear how it is possible to conclude this just from assays of sensitivities.

Response: This conclusion was also supported by the detection of PSIP1/p75, R-loops at gene promoters and gene bodies. Moreover, multiple publications demonstrate the role of PSIP1 in transcription elongation, splicing, and DNA repair at transcription units. It is also known that PSIP1 promotes HR at the transcription units PMID: 22773103 (also reviewed in <https://doi.org/10.1016/j.jmb.2017.03.024>)

- Colocalization of γ H2AX and PSIP1 is not evident from Fig. S5 given that PSIP1 staining seems to be pan-nuclear whereas discrete γ H2AX foci are shown.

Response: PSIP1 is known to be a chromatin protein, has been known to be localised throughout the nucleus. Since we are showing the association of PSIP1 to the damage sites using γ -H2AX IP, we have removed this immunofluorescence data from the manuscript.

- Could the differences in 53BP1 and RAD51 foci formation in PSIP1-sh cells induced by CPT or pheomycin be explained by a differential cell cycle distribution pattern caused by PSIP1 depletion?

Response: Since the PSIP1 KD did not cause any significant change in the cell cycle phases (Fig. S5b and c), the cell cycle distribution may not be the reason for this difference in 53BP1 and RAD51 foci formation.

- In Fig. 5B and C authors relate PSIP1 levels in 2 different prostate cancer cell lines with their PARPi sensitivity. However, results show that the cell line with higher PSIP1 levels (LNCaP) is more sensitive to Olaparib than the one with lower levels of PSIP1 (PC-3), suggesting that there is not a direct correlation between those 2 facts (Olaparib sensitivity and PSIP1 levels). If Olaparib sensitivity was caused by reduced levels of PSIP1, PSIP1 overexpression in those cells lines is expected to suppress PARPi sensitivity. Besides, to conclude a role mediated by R-loops, RNH expression should partially revert the phenotype. Without these additional data, the higher PARPi sensitivity may not be related at all with R-loops.

Response: The difference in PARPi sensitivity between PC-3 and LNCaP cells could be due to other factors like their P53 status etc. (LNCaP: p53 wild type; PC3: P53 mutated). To further confirm the role of PSIP1 and R-loop axis in the drug sensitivity, as suggested by the reviewer, we generated PSIP1 KD lines of LNCaP cells and performed additional experiments. PSIP1 depletion lead to the increased sensitivity to PARPi and that was reversed by RNASEH1 overexpression. These results are included in the revised manuscript (Fig. 5h and 5i).

Minor points:

- In general, numbers of replicates for immunofluorescence experiments should be stated in figures legends.

Response: These details are given in the revised manuscript.

- In line 277, Fig. 5G and H should be Fig. 5D and E.

Response: We thank the reviewer for pointing this out and we have corrected this mistake in the revised manuscript

- In lines 126-127, authors referred to the generation of HEK293T and RWPE-1 cells depleted of PSIP1 but Fig S1G and S1I are missing. In legend of fig. 1G the cell line used for shRNA PSIP1 depletion is omitted.

Response: we have corrected this mistake in the revised manuscript

- In Fig. 2 H & S3D details about the treatment of control cells (are transfected with an empty plasmid in order to be comparable to eGFP-RNaseH transfected cells?) and timing for RNaseH expression are missing.

Response: We have now included these details in the revised manuscript.

- In the M&M some details are missing and should be carefully revised. For example, the compositions of some buffers are not detailed (lysis buffer, methylene blue, NP-40 lysis buffer) or the concentration of some reagents (dilution of antibodies for blots, IFs and PLAs, protein-A Dyna beads). In addition, in the clonogenic assay protocol, it might be an error in the number of plated cells or the plate used (1000 cells in a well of a 96-well plate).

Revision: We have included more details in the methods section and errors are corrected.

Reviewer #3 (Remarks to the Author):

PSIP1/LEDGF reduces R-loops at transcription sites to maintain genome integrity. Jayakumar et al The authors identify PSIP1 as an R-loop interacting protein and show that depletion of PSIP1 results in R-loop increases, which is correlated with increased DNA damage. Increased R-loops are also correlated with transcription arrest and TR conflicts which lead to DNA damage. PSIP1 is also suggesting to have a role in HR repair pathway, although the exact mechanism appears unclear. Finally, decreasing PSIP1 levels sensitizes cancer cells to PAPER inhibitors. Throughout the manuscript the authors suggest that PSIP1 has a direct role in regulating R-loops. There is no evidence for this and the experiments performed cannot distinguish between direct vs indirect effects. Most of the experiments are observational and do not give a clear result in terms of how PSIP1 functions. Overall, while some of the observations made are interesting, many are expected, and the new results appear a little under developed for publication. Further experiments to identify a clear mechanism of action of PSIP1 in R-loop regulation will help in making a case for publication of this manuscript.

Response: The authors thank the reviewer for reviewing the manuscript and offering valuable suggestions that really helped in improving the manuscript. We have carried out additional experiments addressing the deficiency pointed out by the reviewer. As indicated by the reviewer, we have carried out many rescue experiments in PSIP1-depleted cells. These results are now included in the revised manuscript and some of the results are given below.

- RNASEH1 overexpression in the PSIP1-KD cells leads to the reversal of R-loop accumulation mediated by PSIP1 depletion.
- R-loop slot blot (Fig. 1i)

- R-loop immunofluorescence (Fig. 1h)
- Overexpression of P75 and P52 isoforms in PSIP1-depleted cells also resulted in the reversal of R-loop accumulation caused by PSIP1 depletion. (Fig. S2b)
- Overexpression of RNASEH1 resulted in the reversal of DNA damage in PSIP1-depleted cells, (reduced γ -H2AX and 53BP1 foci).
- Reversal of γ -H2AX foci (Fig. 2b)
- 53BP1 foci (Fig. 2c)
- 53BP1 western blotting (Fig. 2d)
- RNASEH1 overexpression also resulted in the reversal of the number of R-loop peaks and γ -H2AX peaks seen in the PSIP1-depleted cells by CUT&Tag (Fig. 2e and f).
- RNASEH1 overexpression also resulted in the reversal of the drug sensitivity mediated by the depletion of PSIP1 against olaparib and illudin-S in prostate cancer cells (Fig. 5h and i).

Specific

Comments:

1. Figure 1C – what is a universal nuclease? There is no band in the PSIP1 input lane. Better quality western is needed.

Response: We have used benzonase in our experiments It is a promiscuous endonuclease that attacks and degrades all forms of DNA and RNA (single-stranded, double-stranded, linear and circular) and hence, we called that a universal nuclease. The enrichment of the PSIP1 after the S9.6 IP was high and they gave a very strong signal and hence the input signal is lightly visible. We have given a better-quality blot in the revised manuscript.

2. Figure 1E – no negative controls are shown in PLA experiments. Also it is not clear how these dots are quantified when individual dots are not even visible.

Response: In the revised manuscript, we have included a single antibody control and RNaseH overexpression control in PLA experiments. (Fig 1e, Fig 1h, 2b, 5h and 5i).

PLA data for S9.6 and PSIP1 with single antibody controls and reversal by RNASEH overexpression (Fig. 1e)

PLA data for S9.6 and γ -H2AX with single antibody controls (Fig. 2g)

PLA data for PCNA and RNAPII with single antibody controls and reversal by RNASEH overexpression (Fig. 3f)

The PLA images are grabbed using an InCell high-content imaging system and analysed using InCarta software attached to the machine. The PLA foci in the nucleus are counted, and the data from 3000- 5000 cells is analysed and plotted.

3. Lines 96-98 – Just because R-loop levels are restored to normal when PSIP1 is reintroduced into cells do not mean this protein has a direct role in reducing R-loop levels. This is an overstatement without any supporting evidence.

Response: We have carried out additional experiments to show PSIP1 could bind to the R-loops (Fig. 1C). And also the R-loops that are accumulated can be reversed by overexpression of RNASEH1 overexpression (Fig. 1h and i, S2a). These data are now included in the revised manuscript and we have also modified the statement.

RNASEH1 overexpression in the PSIP1-KD cells leads to the reversal of R-loop accumulation mediated by PSIP1 depletion. R-loop slot blot (Fig. 1i) and R-loop immunofluorescence (Fig. 1h)

4. The Chedin group has shown that the S9.6 antibody has significant cross reactivity with dsRNA in immunofluorescence experiments (PMID: 33830170). Therefore, the use of this antibody without the appropriate controls in IF experiments is not scientifically sound (Figure 1I).

Response: As suggested by the reviewer, we have employed additional control while detecting R-loops with S9.6 antibody. In the immunofluorescence experiment, we have overexpressed RNASEH1 in the cells and seen the reversal of R-loops accumulation upon the PSIP1 depletion (Fig. 1h, S2a). To validate the specificity of the detection of R-loops by S9.6, we treated the genomic DNA with RNASEH, and it led to the complete disappearance of R-loop bands in the slot blot, indicating the specificity of our detection of R-loops by S9.6 (Fig. S1c, S1F and S1I). Secondly, overexpressing RNASEH1 led to the reversal of R-loop accumulation mediated by PSIP1 depletion. These data are included in the revised manuscript.

R-loop immunofluorescence and its reversal by RNASEH overexpression (Fig. 1h)

To verify the specificity of the S9.6 signal, as suggested by the reviewer, we performed RNASEH control data for MEF cells (Fig. S1c), RWPE-1 cells (Fig. S1f) and HEK293T cells (Fig. S1i). We have included this data in the revised manuscript.

5. Lines 142-143 - It is unclear how knock down experiments can distinguish between direct and indirect effects. While R-loops can increase upon PSIP1 KD, this can happen through effects on transcription elongation, splicing etc. In addition, there is no study that has comprehensively identified all genes involved in R-loop homeostasis. Where do the authors find the 186 genes that they report in Fig S2E. And just because there is no change in these genes does not mean that PSIP1 is the only factor that contributes to aberrant R-loop accumulation in these experiments. This is a major overstatement.

Response: As suggested by the author, we have now modified the statement. The list of 186 genes are pooled from the available literature on R-loop resolution factors. This list of factors and the relevant publication is tabulated and the same is now included in the supplementary information (Supplementary table:1)

6. *Fig 2E and 2F – PSIP1 levels in control cells at sites of h2AX accumulation look barely above background. These experiments do not look conclusive and must be repeated.*

Response: As suggested by the reviewer, we have repeated this experiment in HEK293T cells and found more γ H2AX peaks upon PSIP1 depletion and they got reversed upon overexpression of RNASEH1. These data is included in the revised manuscript.

7. *Figure 2G – This figure has the same quantification issue as Figure 1E.*

Response: We have quantified the dots using InCarta software attached to the InCell high-content imaging system. The number of PLA dots in the nucleus region is analysed from 3000-5000 cells and plotted (detailed in the revised manuscript).

8. *Figure 3 – None of the results presented in this figure are surprising i.e it is well established that r-loops are enriched at regions with high GC skew. It is also known that R-loop accumulation can cause transcriptional arrest and TR conflict. There is no reason to believe that r-loops that accumulate because of PSIP1 depletion would be different. However, if they were different, it would be interesting to report. The data in this figure belongs in the supplement.*

Response: As suggested by the reviewer, we have moved GC skew analysis to the supplement (Fig. S4). The reviewer is right in saying that R-loop accumulation is known to cause transcription replication conflicts. But the role of PSIP1 in R-loop resolution and in the absence of it can lead to transcription replication conflict is not known and we think that is still an interesting and novel finding.

9. *Figure 4 – The authors show that PSIP1 KD cells treated with illudin and etoposide show sustained DNA damage and conclude that PSIP1 has a role in DNA repair. What is the mechanism for this? How exactly does PSIP1 function in DNA repair?*

Response: In a previous study by Duggard et al. PMID: 22773103 (reviewed in <https://doi.org/10.1016/j.jmb.2017.03.024>) has demonstrated the role of PSIP1 in promoting the HR pathway by recruiting CtIP to the damaged sites. Our findings in this study complement Duggard et al. study, which shows the role of PSIP1 in promoting HR at the actively transcribing gene bodies by preventing the accumulation of unscheduled R-loops during transcription and also helping in the repair of the damage caused by the R-loops.

10. *Figure 4C – PSIP is pulled down with gH2AX and this is increased with CPT treatment that increases R-loops. This increase should be shown because the timing of CPT treatment has differential effects on R-loops. This result is a little confusing because R-loops are supposed to be nucleosome-depleted regions. So does this imply that PSIP1 binds in the vicinity of R-loops and through gH2AX?*

Response: We treated the cells with CPT for 2 h to induce R-loop mediated DNA damage (PMID: 34049076). We have verified the ability of CPT to cause R-loop accumulation (after 1 h) and DNA damage (after 2 h). We have included this data in

the revised manuscript (Fig. S6d and e). It is tricky to infer the dependency and order of recruitment of gH2AX, PSIP and R-loop as it is an IP of the endogenous complex.

It is tricky to infer the dependency and order of recruitment of gH2AX, PSIP and R-loop as it is an IP of endogenous complex, our data suggest that R-loop, PSIP1 and gH2AX.

11. Figure 5 – If PSIP1 levels are what sensitizes cancer cells to PARP inhibitors, does PSIP1 KD in RWPE1 cells have the same effect?

Response: To further confirm the role of PSIP1 and R-loop axis in the drug sensitivity, we generated PSIP1 KD lines of LNCaP cells and performed additional experiments. PSIP1 depletion led to the increased sensitivity to PARPi and that was reversed by RNASEH1 overexpression. These results are included in the revised manuscript (Fig. 5h-i).

REVIEWER COMMENTS

Reviewer #1 (Remarks to the Author):

The authors have addressed my original concerns appropriately. I now support publication.

Reviewer #2 (Remarks to the Author):

The manuscript has improved greatly and provides more convincing data. The authors have incorporated several additional controls to ensure the specificity of the signals and the suppression of some data using the overexpression of RNaseH1. These controls effectively demonstrate the dependence of those phenotypes on the formation of DNA-RNA hybrids and the impact of PSIP1 for suppressing R-loops accumulation and R-loop-mediated DNA damage. Nevertheless, some aspects of the manuscript still needs attention.

- The CUT&TAG experiments of PSIP1, S9.6, and gammaH2AX are intended to demonstrate a direct role of PSIP1 on R-loops and the prevention of R-loop-dependent DNA damage by relating PSIP1 binding sites to R-loop and gammaH2AX new peaks arising in PSIP1 KD cells. The authors have introduced now a novel upSet plot in Figure 3c, demonstrating the intersection of PSIP1, S9.6, and gammaH2AX peaks from the CUT&TAG experiments. However, the data categorization within each section is somewhat challenging to follow. A more informative approach, such as employing Venn diagrams and providing a description indicating the percentage of gammaH2AX and S9.6 peaks that are gained upon PSIP1 depletion and coincide with PSIP1 peaks in control cells (out of the total S9.6 peaks detected upon PSIP1 depletion), would aid readers to understand the analysis.

CUT&TAG experiments show enrichment of R-loops in PSIP1 KD cells and PSIP1 around gene promoters and authors suggest that PSIP1 might reduce R-loops at G-rich promoters with high GC skew. However, data in figure S4 show a similar binding of PSIP1 at regions with positive and negative GC skew and a similar response of R-loops and gammaH2AX enrichment upon PSIP1 KD in both types of regions. Thus, G-rich promoters show a high accumulation of PSIP1 and R-loops upon PSIP1 depletion but it seems that gamma-H2AX does not exhibit a dramatic accumulation in these regions compared to others within the KD cells. Clarify this.

- The available data from figure 3d and S5 is not sufficient to conclusively establish that R-loops in PSIP1 KD cells are the causative factor for the observed transcription defects in the absence of PSIP1 due to collision with the replication fork. It could be due to a more direct role of PSIP mutation on transcription. A comparison using TT-seq between PSIP1 KD-specific R-loop regions and PSIP1 KD non-specific R-loop regions would confirm the specificity of the transcription defect in R-loop regions.

- With respect to the suggestion for the direct detection of DNA-RNA hybrids by DRIP-qPCR at PSIP1- and γ H2AX-enriched genomic regions, the authors indicate that "repeating PCR for qPCR is not ideal after a CUT&TAG experiment." Nevertheless, DRIP-qPCR could be conducted independently of the CUT&TAG experiment.

Minor points:

In Figure 1c, it is essential to incorporate quantification of the signals relative to the loading DNA. Additionally, ensure uniform spacing between signals in the top left panel and the rest of the panels.

In the S9.6-PSIP PLA experiment depicted in Figure 1e, the authors have included quantification of the foci observed in single S9.6 antibody reactions, yet the quantification of single PSIP1 antibody reactions is absent. The Y-axis label in Figure 1e is absent.

The quantification of 53BP1 foci in Figure 2c is missing.

In Figure 2d (western blot of 53BP1), the signal of 53BP1 is reduced to a comparable extent upon RNase H1 overexpression in control cells, as it is in PSIP1-depleted cells. Hence, it would be difficult to conclude that there is a suppression of the phenotype upon RNaseH1 overexpression.

The numbers on the Y-axis in Figure 1f require correction. Furthermore, it is needed to specify the statistical test utilized.

In Figure 2f, expanding the coverage of the genome region to encompass signals from various genes and including the coordinates of the genomic region displayed (from x to y) would enhance the quality of the analysis.

Please, clarify the nature of the white signals in the PSIP1 heatmap depicted in Figure 3b and include if the analysis correspond to control or shPSIP1 cells in the figure legend.

Reviewer #3 (Remarks to the Author):

The authors have addressed all the concerns I had in the first round of review and this manuscript is now suitable for publication.

Reviewer #2 (Remarks to the Author)

The manuscript has improved greatly and provides more convincing data. The authors have incorporated several additional controls to ensure the specificity of the signals and the suppression of some data using the overexpression of RNaseH1. These controls effectively demonstrate the dependence of those phenotypes on the formation of DNA-RNA hybrids and the impact of PSIP1 for suppressing R-loops accumulation and R-loop-mediated DNA damage. Nevertheless, some aspects of the manuscript still needs attention.

Response: The authors thank the reviewer for providing with us the comments for improving the manuscript. Our point-by-point response to the comments is given below.

- The CUT&TAG experiments of PSIP1, S9.6, and gammaH2AX are intended to demonstrate a direct role of PSIP1 on R-loops and the prevention of R-loop-dependent DNA damage by relating PSIP1 binding sites to R-loop and gammaH2AX new peaks arising in PSIP1 KD cells. The authors have introduced now a novel upSet plot in Figure 3c, demonstrating the intersection of PSIP1, S9.6, and gammaH2AX peaks from the CUT&TAG experiments. However, the data categorization within each section is somewhat challenging to follow. A more informative approach, such as employing Venn diagrams and providing a description indicating the percentage of gammaH2AX and S9.6 peaks that are gained upon PSIP1 depletion and coincide with PSIP1 peaks in control cells (out of the total S9.6 peaks detected upon PSIP1 depletion), would aid readers to understand the analysis.

Response: As suggested by the reviewer, a subset of data from the upset plot (PSIP1 peaks in wild type cells, S9.6 peaks in PSIP1-KD cells and γ -H2AX peaks from PSIP1-KD) along with the overlapping peaks have been represented as Venn diagram for easy understanding (Fig. 3c).

CUT&TAG experiments show enrichment of R-loops in PSIP1 KD cells and PSIP1 around gene promoters and authors suggest that PSIP1 might reduce R-loops at G-rich promoters with high GC skew. However, data in figure S4 show a similar binding of PSIP1 at regions with positive and negative GC skew and a similar response of R-loops and gammaH2AX enrichment upon PSIP1 KD in both types of regions. Thus, G-rich promoters show a high accumulation of PSIP1 and R-loops upon PSIP1 depletion but it seems that gamma-H2AX does not exhibit a dramatic accumulation in these regions compared to others within the KD cells. Clarify this.

Response: We thank the reviewer for pointing this out. In the CUT&Tag peak analysis, we found that upon PSIP1 depletion, there was almost two-fold increase in the number of γ -H2AX peaks in the promoter regions (Fig. 3a). But at the G-rich promoter regions, we don't see the proportional increase in the number of γ -H2AX peaks upon PSIP1 depletion (Fig. S4a). We were also intrigued by this and currently we don't have a clear mechanistic detail. It could be due to the other factors playing role in that context and not allowing the γ -H2AX to accumulate.

- The available data from figure 3d and S5 is not sufficient to conclusively establish that R-loops in PSIP1 KD cells are the causative factor for the observed transcription defects in the

absence of PSIP1 due to collision with the replication fork. It could be due to a more direct role of PSIP mutation on transcription. A comparison using TT-seq between PSIP1 KD-specific R-loop regions and PSIP1 KD non-specific R-loop regions would confirm the specificity of the transcription defect in R-loop regions.

Response: We agree with the reviewer that we can not rule out the direct effect of PSIP depletion on reduced transcription. We have added this as a limitation in the revised manuscript (page 6 first sentence)

- With respect to the suggestion for the direct detection of DNA-RNA hybrids by DRIP-qPCR at PSIP1- and γ H2AX-enriched genomic regions, the authors indicate that "repeating PCR for qPCR is not ideal after a CUT&TAG experiment." Nevertheless, DRIP-qPCR could be conducted independently of the CUT&TAG experiment.

Response: Since we do not have expertise in DRIP-qPCR experiments, we had to prioritise other experiments for this revision. Moreover, recent study comparing the efficacy of different methods to quantify the R-loops has shown that CUT&Tag is an equally effective method for detecting and estimating the R-loops compared to DRIP-seq (Wang et al., Science Advances; PMID: 33597247). Hence, we have used CUT&Tag for our study, and the detection specificity of CUT&Tag has been verified by RNASEH reversal. Though we value the suggestion of the reviewer, we are convinced about the CUT&Tag for detecting the R-loops and hence we did not include DRIP-qPCR in the current study.

Minor points:

In Figure 1c, it is essential to incorporate quantification of the signals relative to the loading DNA. Additionally, ensure uniform spacing between signals in the top left panel and the rest of the panels.

Response: As suggested by the reviewer, we have quantified the band intensity and the graph is included in the revised manuscript (Fig. 1c). We have also adjusted the spacing between the lanes to look uniform (Fig. 1c)

In the S9.6-PSIP PLA experiment depicted in Figure 1e, the authors have included quantification of the foci observed in single S9.6 antibody reactions, yet the quantification of single PSIP1 antibody reactions is absent. The Y-axis label in Figure 1e is absent.

Response: We thank the authors for pointing this out. We have included the PSIP1 single antibody control and the graph in the Fig. 1e has been accordingly modified. The Y-axis is also labelled now (Fig. 1e).

The quantification of 53BP1 foci in Figure 2c is missing.

Response: As suggested by the reviewer, the quantification of 53BP1 foci has been performed from the images obtained from the InCell high-content imaging system and the same is plotted and included in the revised manuscript (Fig. 2c).

In Figure 2d (western blot of 53BP1), the signal of 53BP1 is reduced to a comparable extent

upon RNase H1 overexpression in control cells, as it is in PSIP1-depleted cells. Hence, it would be difficult to conclude that there is a suppression of the phenotype upon RNaseH1 overexpression.

Response: The reduction in the 53BP1 levels upon RNASEH overexpression in the control cells could be due to the difference in loading. We have included a different blot in the revised manuscript (Fig. 2d).

The numbers on the Y-axis in Figure 1f require correction. Furthermore, it is needed to specify the statistical test utilized.

Response: We thank the reviewer for pointing out this and have corrected the Y-axis in the revised manuscript (Fig. 1f). We have also included the statistical analysis in the figure legend.

In Figure 2f, expanding the coverage of the genome region to encompass signals from various genes and including the coordinates of the genomic region displayed (from x to y) would enhance the quality of the analysis.

Response: We have now revised the figure

Please, clarify the nature of the white signals in the PSIP1 heatmap depicted in Figure 3b and include if the analysis correspond to control or shPSIP1 cells in the figure legend.

Response: The heatmap in Figure 3b was generated by calculating Log₂ WT/PSIP1-KD and the scale is given below the heatmap. In the PSIP1 heatmap, the yellow signals mean that the presence of peak signal in the WT cells that are lost upon knockdown. The details in the legends are modified in the revised manuscript.